# Overclocking LLM Reasoning: Monitoring and Controlling Thinking Path Lengths in LLMs

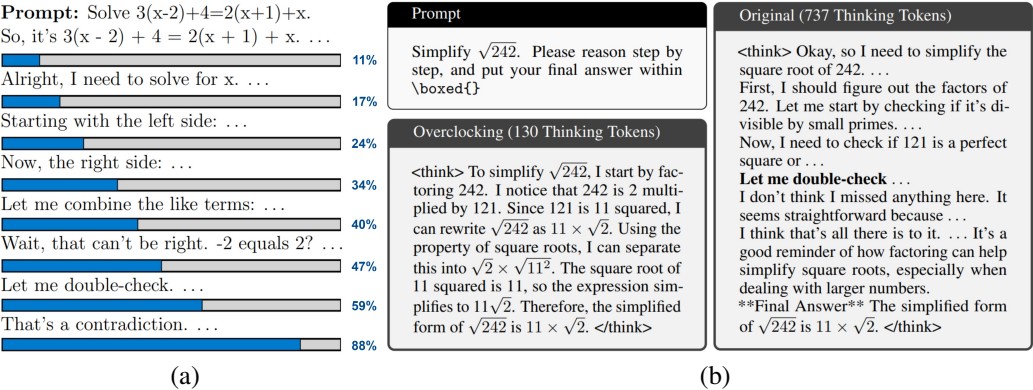

Figure 1: Applications of our method. (a) Monitoring the reasoning progress. (b) Overclocking it.

## ABSTRACT

Recently, techniques such as explicit structured reasoning have demonstrated strong test-time scaling behavior by enforcing a separation between the model's internal "thinking" process and the final response. A key factor influencing answer quality in this setting is the length of the thinking stage. When the reasoning is too short, the model may fail to capture the complexity of the task. Conversely, when it is too long, the model may overthink, leading to unnecessary computation and degraded performance. This paper explores and exploits the underlying mechanisms by which LLMs understand and regulate the length of their reasoning during explicit thought processes. First, we show that LLMs encode their progress through the reasoning process and introduce an interactive progress bar visualization, which is then used to reveal insights on the model's planning dynamics. Second, we manipulate the internal progress encoding during inference to reduce unnecessary steps and generate a more concise and decisive chain of thoughts. Our empirical results demonstrate that this "overclocking" method mitigates overthinking, improves answer accuracy, and reduces inference latency. Our code is publicly available.

## 1 INTRODUCTION

In recent years, Large Language Models (LLMs) such as ChatGPT (OpenAI, 2023) have demonstrated remarkable capabilities across both general-purpose and domain-specific challenges, exhibiting surprising generalization abilities (Kojima et al., 2022). An emerging frontier for enhancing their performance is test-time scaling, which aims to improve model responses by dynamically allocating additional computational effort during inference (Snell et al., 2024; Welleck et al., 2024).

A prominent strategy for enabling effective long-form reasoning is explicit structured reasoning, which separates the model's reasoning process from its final answer (Wei et al., 2022). This is commonly implemented using special tokens that mark the start and end of the reasoning phase (e.g., <think> and </think>, which are employed by DeepSeek-R1 (Guo et al., 2025)), encouraging the model to deliberate before responding.

The effectiveness of such techniques depends heavily on the length of the thinking stage (Jin et al., 2024a; Wu et al., 2025b). Too few steps may fail to capture sufficient task complexity, while too many can lead to overthinking and unnecessary computation (Sui et al., 2025). Thus, controlling the thinking path length and understanding its underlying mechanisms are essential for balancing accuracy, efficiency, and responsiveness.

While several prior works have explored the impact of reasoning length on model quality and proposed techniques to control or optimize response length through specialized training procedures (Aggarwal & Welleck, 2025) or prompting (Wu et al., 2025a), they do not investigate the underlying mechanistic design that governs how reasoning length is determined. Moreover, none of these approaches address the unique case of structured reasoning, where the model explicitly separates a holistic thinking stage from the final response. We consider this setting particularly important, as it offers a valuable opportunity to better understand and exploit the "thinking phase" of reasoning models.

This work tackles both of these gaps. First, we investigate whether LLMs are capable of monitoring their relative position within the thinking process. Possessing even an implicit form of progress monitoring may enable aspects of key cognitive concepts such as self-regulation (Zimmerman, 2002) and metacognition (Flavell, 1979). To identify mechanisms that encode this information, we perform a regression analysis, and show that the relative position can be captured by projections that we term "progress vectors". The extracted information is then used to create an interactive loading bar visualization, see Figure 1(a) that depicts the model's progress throughout the thinking phase, making the reasoning process more transparent and easier for users to collaborate with.

The ability to extract progress information does not mean that the model employs it mechanistically, unless an intervention analysis is performed. We thus manipulate the internal representation along the progress vectors and achieve a clear modulation of the length of the thinking phase, showing overclocking effects. The former is depicted in Figure 1(b). Reassuringly, this modulation does not tend to be detrimental to the LLM's performance. In fact, we show that overclocking can improve the model's performance by mitigating overthinking, enhancing computational efficiency, and tailoring the model's reasoning depth to each task's complexity.

**Our main contributions are as follows:** (i) We provide the first empirical evidence that LLMs maintain an internal estimate of their relative position within the explicit thinking phase. This finding sheds light on the plausibility of planning and self-monitoring abilities of LLMs, concepts typically associated with cognitive capabilities. (ii) We identify an internal encoding of this information by learning progress vector projections, and employ these projections to expose a dynamic thinking progress bar. (iii) We perform an intervention study and manipulate the progress vectors to overclock and downclock the reasoning process. Finally, (iv) We empirically demonstrate that interventions of the progress vectors improve both the efficiency and the effectiveness of strong LLMs such as DeepSeek-R1 by mitigating overthinking and reducing the generation of unnecessary reasoning steps.

## 2  RELATED WORK

**Test-time scaling** refers to a recent trend in which the reasoning depth of an LLM is dynamically increased during inference, allowing the model to perform complex multi-step reasoning and extend its problem-solving capabilities (Snell et al., 2024; Welleck et al., 2024). This approach heavily relies on the model's ability to reason effectively over long trajectories. One notable technique for improving the effectiveness of multi-step reasoning is Chain-of-Thought (CoT) prompting (Wei et al., 2022), which encourages the model to generate a sequence of intermediate reasoning steps before producing its final answer by incorporating additional guidance into the prompt. This technique has been adopted and extended in models such as DeepSeek-R1 (Guo et al., 2025), OpenAI's O1 (Jaech et al., 2024), and S1 (Muennighoff et al., 2025) which are fine-tuned on datasets containing CoT-style reasoning steps prior to the final answer. A further refinement was proposed by DeepSeek-R1 (Guo et al., 2025), which explicitly enforces open-ended structured reasoning through the use of dedicated <think> and </think> tokens that mark the beginning and end of the model's "thinking" phase.

**The length of the reasoning chain** is a key factor influencing the performance of modern reasoning LLMs. If the chain is too long, the LLM may engage in unnecessary computation and suffer from overthinking (Sui et al., 2025; Su et al., 2025; Chen et al., 2024), a phenomenon in which excessive reasoning steps degrade answer quality. On the other hand, if the reasoning is too short, the LLM may fail to capture the complexity required to solve the task effectively (Wang et al., 2025). To address this trade-off, SoTA LLMs are increasingly designed to support adaptive thinking, where the length of the reasoning phase is dynamically adjusted based on task complexity. Two main strategies for improving the length-control the duration of a model's reasoning have been proposed: (i) Inference-time methods modify the prompt or inject guidance while the model is generating its chain of thought. Notable examples include Kojima et al. (2022), who optimize the initial prompt, and Wu et al. (2025a); Jin et al. (2024b), who dynamically intervene during reasoning by prompting to shorten or lengthen the

reasoning trajectory (ii) Training-time methods fine-tune the model with objectives and datasets that explicitly encourage regulation of the reasoning length. A notable example is Length-Controlled Policy Optimization (LCPO) (Aggarwal & Welleck, 2025), which frames reasoning-length control as a RL objective. Beyond the context of controlling reasoning-chain length, length control has long been studied in standard LLMs to manage overall output length. A substantial body of work addresses this objective across applications such as summarization (Kwon et al., 2023; Retkowski & Waibel, 2024). In contrast to these methods that primarily aim to control the length of the generated answer for efficiency or user-specific preferences, our work focuses on the thinking length rather than the final output. This allows us to gain insights into the model's planning, monitoring, and control capabilities, which we then leverage to improve model quality and model responsiveness.

**Mechanistic interpretability** aims to reverse-engineer LLMs so they can be understood, trusted, and controlled (Elhage et al., 2021; Olah et al., 2020). This is done by connecting observable behaviors to concrete neural components by assigning semantic roles to individual neurons or weights and isolating compact "circuits". Prominent examples include the induction-heads (Olsson et al., 2022) and mechanisms for arithmetic (Kantamneni & Tegmark, 2025; Nanda et al., 2023; Zhong et al., 2023; Chughtai et al., 2023) and in-context learning via task-vectors (Hendel et al., 2023).Our work follows this line of research and aims to uncover the intrinsic mechanisms that encode a model's relative position within its internal reasoning process, with the goal of better understanding and even controlling the planning abilities of reasoning LLMs.

## 3 METHOD

We investigate monitoring and control of the thinking phase in reasoning LLMs. In Sec. 3.1, we show that the model's hidden representations encode information about its position within the thinking phase. This allows for real-time monitoring in the form of a loading bar and reveals distinct thinking patterns. In Sec. 3.2, we show that this mechanism can be adjusted to control thinking length.

### 3.1 MONITORING THE THINKING-PHASE

We hypothesize that as part of learning to reason effectively, models must also implicitly learn to track their progress through the thinking phase, maintaining an estimate, for example, of how close they are to the final answer. Since progress-tracking is input-dependent, such information cannot be stored in the model's static weights and must instead be encoded dynamically in the hidden representations passed between layers. We chose to extract information from the final hidden layer for two main reasons. First, this layer provides the richest token-level representations, having integrated information from all previous layers. Second, our goal is to eventually intervene in the reasoning process by manipulating the thinking progress. The final hidden layer is especially suitable for this because it is close to the token embedding space, allowing for effective intervention. Moreover, it lies downstream of the attention layers, which cache intermediate states. This means that modifying the hidden representation at this stage can influence the immediate output token without disrupting the model's internal memory, thereby isolating the effect of the intervention to a single step.

We focus on models that perform explicit structured reasoning, characterized by a reasoning phase delimited by <think> and </think> tokens, as employed in DeepSeek-R1 (Guo et al., 2025). This enables us to quantify the model's progression within the thinking phase by precisely labeling each token with an interpolated value between zero and one, according to its relative position.

To learn a thinking progress predictor, we initially sample entries from a dataset of mathematical problems. We then prompt a reasoning model with a single problem each time, generating an answer that contains a thinking phase. For a generation that ended successfully, we specifically observe its thinking trajectory $T = w_1 w_2 \ldots w_N$, which is the sequence of tokens between "<think>" and "</think>", and collect the representations from the last layer, $\{\mathbf{h}_j^{(k)} \in \mathbb{R}^d \mid j = 1, \ldots, N_k\}$ for each token $j$ in the $k$ trajectory, where $d$ is the hidden size of the reasoning model and $N_k$ is the number of tokens in that trajectory. We pair each hidden state with its normalized position in the sequence as a regression label. Note that layers other than the last one can also serve as a basis for our TPV monitoring, see App. C. However, for intervention, we observe that targeting the last layer is optimal, probably because it minimizes the distribution shift induced by the intervention.

Formally, we construct a dataset $\mathcal{D} = \{(\mathbf{h}_j^{(k)}, p_j^{(k)}) \mid k = 1, \ldots, K, \ j = 1, \ldots, N_k\}$, where $\mathbf{h}_j^{(k)} \in \mathbb{R}^d$ is the hidden representation of the $j$-th token in the $k$-th thinking trajectory, and $p_j^{(k)} = \frac{j}{N_k} \in (0, 1]$ is the token's relative position in its sequence, where $K$ denotes the number of trajectories.

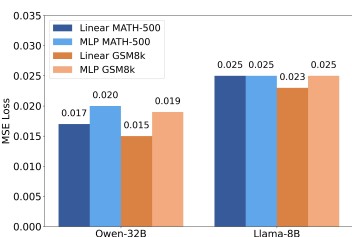
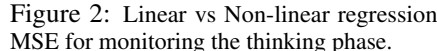

Figure 2: Linear vs Non-linear regression MSE for monitoring the thinking phase.

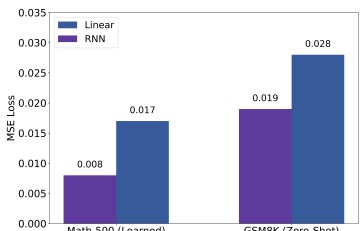

Figure 3: MSE of Linear vs RNN in learned and zero-shot settings.

This allows us to optimize for a progress extraction function $f_\theta$ which maps the hidden representations to their relative positions in the form of a regression task $\theta^* = \arg\min_\theta \sum_{(\mathbf{h},p)\in\mathcal{D}} (f_\theta(\mathbf{h}) - p)^2$.

We conduct experiments using two reasoning models: DeepSeek-R1-Distill-Qwen-32B and DeepSeek-R1-Distill-Llama-8B (Guo et al., 2025). Both are designed to explicitly separate the thinking process from final answers. We evaluate them on two benchmark datasets of mathematical problems: Math500 (Lightman et al., 2023) and GSM8k (Cobbe et al., 2021). For each dataset, we sample 30 problems and prompt each model to generate 5 distinct responses per problem using a temperature of 0.6. The prompt format includes the instruction: "Please reason step by step, and put your final answer within \boxed{}. <think>", as recommended by Guo et al. (2025). Following the dataset curation procedure described earlier, we obtain four distinct progress regression datasets, each corresponding to a unique combination of reasoning model and problem set. We split each dataset into train and test sets using an approximate 80/20 ratio based on entire thinking trajectories, ensuring that all generations for a given problem remain within the same split.

We first fit a linear regressor parametrized by $\boldsymbol{\theta} \in \mathbb{R}^d$ as the function $f_\theta$ for estimating the progress property $\bar{p}_j^{(i)} = \boldsymbol{\theta}^T \mathbf{h}_j^{(i)}$. We refer to the parameter vector $\boldsymbol{\theta}$ as the *"thinking progress vector"* (**TPV**).

To assess whether the estimated progress $\bar{p}_j^{(i)}$ can be more effectively captured by a more complex model, we compare a simple two-layer feedforward network (FFN) with our proposed TPV model. As shown in Figure 2, both models achieve low loss on the test sets, with no observable improvement from the FFN. Guided by the principle of favoring simplicity, and supported by these empirical findings, we select TPV as our method of choice for progress extraction in subsequent experiments.

To improve the predictions, we leverage the model's autoregressive nature, in which tokens are generated sequentially, and apply exponential smoothing over the prediction history to reduce noise. In Figure 5, we illustrate TPV predictions over problems from the Math-500 test set (Lightman et al., 2023). The light blue and dark blue points represent the predictions and smoothed predictions correspondingly, while the orange line depicts the ideal loading bar obtained through linear interpolation. Figure 5(a) presents an aggregative view over data points from several thinking trajectories, while Figures 5 (b, c) showcase TPV predictions and smoothed predictions along thinking trajectories of single problems in the Math-500 test set. As shown in the figures, both approaches, with and without smoothing, successfully predict the relative position, while the latter produces more precise results that can be used to create a clearer, more interpretable loading bar.

Motivated by this observation, and to better exploit the temporal structure of the progress-bar prediction task, we replace exponential smoothing with a trainable sequence model, and adopt a single-layer GRU, parameterized by $\theta_{\text{RNN}}$, whose hidden and input dimensions are both set to $d$, and whose output at each step is a scalar. The RNN is trained on the sequence dataset $\mathcal{D}'$, using the same training samples as $\mathcal{D}$, only with a sequence of relative positions rather than performing single-step predictions: $\mathcal{D}' = \{ (\mathbf{h}_1^{(k)}, \ldots, \mathbf{h}_{N_k}^{(k)}), (p_1^{(k)}, \ldots, p_{N_k}^{(k)}) \mid k = 1, \ldots, K \}$. To extend our analysis, we train the model on the MATH-500 training set and evaluate it on both the MATH-500 and GSM8K test sets, thereby covering both the learned and zero-shot regimes. The results are presented in Figure 3, showing that the RNN consistently achieves lower loss than the linear model in both settings. Although performance drops slightly in the zero-shot regime, the loss remains relatively low, suggesting that the model generalizes well across datasets. Finally, Figure 4 presents a qualitative analysis of the RNN's predictions over time during the thinking phase. In both learned and zero-shot regimes, the RNN predictions (shown in purple) produce outputs that are smooth, consistent over time steps, monotonically increasing, and accurately reflect the progress of the thinking phase, especially when combined with additional exponential smoothing.

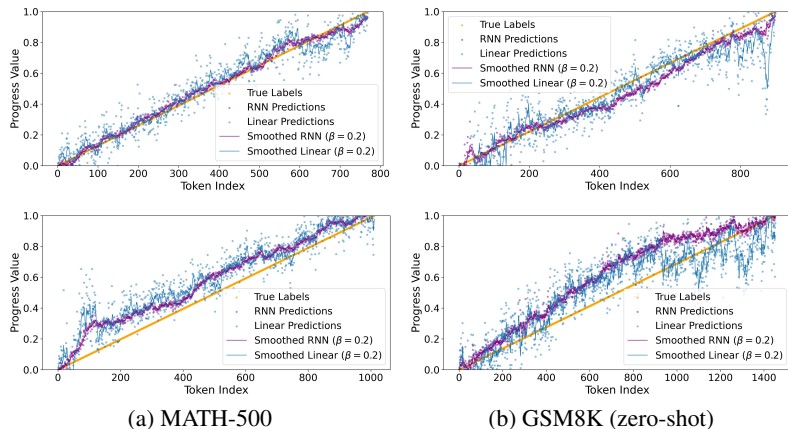

(a) MATH-500      (b) GSM8K (zero-shot)

Figure 4: **Qualitative analysis of RNN-based progress prediction:** Predictions across time steps during the thinking phase. (a) Results on MATH-500 (in-domain); (b) results on GSM8K (zero-shot).

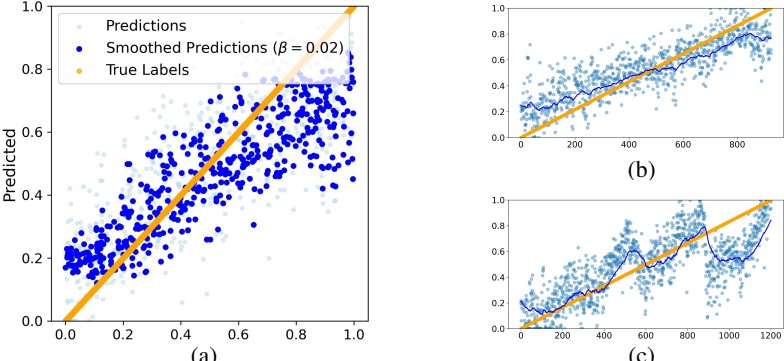

(a)      (b)      (c)

Figure 5: **Relative Progress Prediction Analysis:** (a) We sample 1000 random tokens from the thinking trajectories across all examples in the test set of the Math-500 dataset. The x-axis denotes the ground-truth relative position within the thinking phase, while the y-axis shows the predicted progress. (b, c) Two representative examples from the Math500 test set, illustrating how the predictor estimates progress across the full thinking trajectory of a single generation. The x-axis denotes the decoding step index, and the y-axis shows the predicted value in the $[0, 1]$ range.

## 3.2 Controlling Thinking Progress

A crucial question that arises is whether TPVs reflect a fundamental mechanism that the model uses to track its reasoning progress, or if they are merely residual artifacts that correlate with progress but do not play a causal role in the computation. To address this, we conduct an intervention, which shifts the hidden representation $\mathbf{h}$ in the direction of projection vector $\boldsymbol{\theta}$ by an amount $\alpha$: $\mathbf{h}_\alpha = \mathbf{h} + \alpha\boldsymbol{\theta}$. The altered representation has a new prediction value: $\boldsymbol{\theta}^T\mathbf{h}_\alpha = \boldsymbol{\theta}^T(\mathbf{h} + \alpha\boldsymbol{\theta}) = \bar{p} + \alpha||\boldsymbol{\theta}||^2$.

By performing this intervention past all attention layers, we intervene with next token prediction and refrain from editing representation values that are cached and used in consecutive decoding steps. This way, next token predictions along the thinking trajectory are influenced in two ways: (i) locally, by the intervention-edited representation values; (ii) and historically, derived from the auto-regressive process, through the tokens that were already potentially altered as a result of previous interventions.

In our experiments, we treat $\alpha$ as a hyperparameter that determines the strength of the intervention. Setting $\alpha = 0$ results in no intervention, preserving the original computation. Positive values of $\alpha$ induce *overclocking*. We hypothesize that overclocking will accelerate the model's reasoning phase, making it shorter and more decisive.

## 4 Experiments

This section presents empirical evaluation for both components of our method: (i) TPV-based monitoring, as introduced in Section 3.1, and (ii) the intervention technique described in Sec. 3.2, which overclocks the thinking phase of reasoning LLMs.

Table 1: TPV monitoring performance and training/validation sizes (in thousands of data points) for DEEPSEEK-R1-DISTILL-QWEN-32B (Q32B) and DEEPSEEK-R1-DISTILL-LLAMA-8B (L8B).

| Dataset | Train/Val (k) | | MSE | | $R^2$ | |
|---|---|---|---|---|---|---|
| | Q32B | L8B | Q32B | L8B | Q32B | L8B |
| Math500 | 21.3/7.9 | 281.2/7.9 | 0.017 | 0.025 | 0.65 | 0.65 |
| GSM8K | 20.3/4.9 | 47.4/16.0 | 0.015 | 0.023 | 0.74 | 0.72 |
| HumanEval | 149.8/39.4 | 143.4/49.5 | 0.025 | 0.038 | 0.75 | 0.67 |
| StrategyQA | 62.9/16.1 | 54.6/13.0 | 0.029 | 0.030 | 0.68 | 0.65 |
| GPQA | 479.6/128.1 | 404.1/163.2 | 0.035 | 0.032 | 0.61 | 0.65 |

Table 2: **Results for DeepSeek-R1-Qwen-32B** on GSM-8K and Math500 at different response lengths (512, 1024, 2048 for Math500; 256, 512, 1024 for GSM-8K), showing #Correct '#Cr', #Answered '#An' and #Ended '#En' for each.

| Method | Math500 | | | | | | | | | GSM-8K | | | | | | | | |
|---|---|---|---|---|---|---|---|---|---|---|---|---|---|---|---|---|---|---|
| | 512 | | | 1024 | | | 2048 | | | 256 | | | 512 | | | 1024 | | |
| | #Cr | #An | #En | #Cr | #An | #En | #Cr | #An | #En | #Cr | #An | #En | #Cr | #An | #En | #Cr | #An | #En |
| Base | 36 | 38 | 28 | 154 | 156 | 101 | 278 | 280 | 245 | 12 | 16 | 9 | 221 | 223 | 227 | 285 | 298 | 296 |
| Temperature | 54 | 59 | 43 | 146 | 149 | 119 | 294 | 305 | 294 | 39 | 40 | 33 | 237 | 249 | 248 | 237 | 249 | 300 |
| Instruct | 53 | 59 | 52 | 211 | 214 | 177 | **316** | 321 | 296 | 16 | 18 | 15 | 224 | 237 | 232 | 285 | 299 | 300 |
| Token-Blocking | 19 | 19 | 2 | 174 | 182 | 93 | 298 | 304 | 283 | 14 | 17 | 11 | 195 | 204 | 175 | 276 | 286 | 291 |
| TPV $\alpha = 5$ | 58 | 58 | 42 | 162 | 165 | 98 | 277 | 280 | 257 | 23 | 26 | 14 | 226 | 237 | 233 | 285 | 297 | 293 |
| TPV $\alpha = 100$ | 100 | 106 | 94 | 201 | 213 | 167 | 284 | 296 | 268 | 36 | 41 | 29 | 245 | 256 | 252 | 285 | 299 | 298 |
| TPV $\alpha = 5$ Ins | 123 | 126 | 109 | 240 | 254 | 218 | **316** | 331 | 306 | 34 | 26 | 34 | 239 | 250 | 248 | **286** | 300 | 300 |
| TPV $\alpha = 100$ Ins | **148** | 160 | 142 | **256** | 290 | 279 | 300 | 337 | 330 | **45** | 49 | 40 | **247** | 263 | 261 | 284 | 298 | 298 |

**Monitoring** We evaluate TPV-based monitoring on five datasets using DEEPSEEK-R1-DISTILL-QWEN-32B and DEEPSEEK-R1-DISTILL-LLAMA-8B. We include the two mathematical benchmarks examined earlier in Section 3.1, Math500 and GSM8K, as well as three additional settings: HumanEval for code generation (Chen et al., 2021), StrategyQA for commonsense reasoning (Geva et al., 2021), and GPQA for scientific reasoning (Rein et al., 2023). As shown in Table 1, TPV achieves consistently strong predictive performance across all five datasets, with $R^2$ values ranging from 0.61 to 0.75. These results demonstrate that TPV provides a reliable signal of progress across diverse reasoning tasks.

**Effectiveness of the intervention** We measure the effectiveness of TPVs in Tables. 2, and 3 which report results for DeepSeek-R1-Qwen-32B and DeepSeek-R1-LLaMA-8B respectively. Each table summarizes performance on two mathematical datasets evaluated under three token budgets. The right half of each table presents GSM-8K scores for budgets of 256, 512, and 1024 tokens, whereas the left half presents Math-500 scores for budgets of 512, 1024, and 2048 tokens.

For each regime we report three key metrics to evaluate model performance: (i) #**Correct** - the number of problems for which the model produced a correct final answer. We define the answer as the expression enclosed within the \boxed{} symbol. This pattern is matched programmatically and verified manually. (ii) #**Answered** - the number of generations in which the model produced an answer, operationally determined by the presence of the \boxed{} symbol in the output. (iii) #**Ended** - the number of generations that concluded naturally before hitting the enforced token limit.

It is important to note that the number of answered problems can exceed the number of completed generations. This is because models often produce the \boxed{} token at the end of the thinking phase, before completing the full response, allowing truncated generations to still include a valid answer. The opposite may also occur: particularly the LLaMA model occasionally fails to follow the instruction to wrap the answer in \boxed{}, causing exclusion of otherwise valid answers.

To comprehensively evaluate our method, we benchmark it against the following baselines: (i) **Base Model:** Running the model on the problem concatenated with the instruction "Please reason step by step, and put your final answer within \boxed{}", as defined in DeepSeek-R1 prompting guidelines. (ii) **Temperature:** Same as (i) except we generate five samples and select the shortest answer. Following (Guo et al., 2025), we set the temperature to 0.6. (iii) **Instruct:** A straightforward prompting strategy encourages the model to produce decisive answers with the following prompt: "Please reason step by step, place your final answer inside \boxed{}, and then immediately stop with < |end_of_sentence| >. Present all necessary calculations or arguments concisely, avoiding unnecessary elaboration or verbosity. < think >". **Token-Blocking**: prevent the generation of words

Table 3: **Results for DeepSeek-R1-LLaMA-8B** on GSM-8K and Math500 at different response lengths (512, 1024, 2048 for Math500; 256, 512, 1024 for GSM-8K), showing #Correct '#Cr', #Answered '#An' and #Ended '#En' for each.

| Method | Math500 | | | | | | | | | GSM-8K | | | | | | | | |
|---|---|---|---|---|---|---|---|---|---|---|---|---|---|---|---|---|---|---|
| | 512 | | | 1024 | | | 2048 | | | 256 | | | 512 | | | 1024 | | |
| | #Cr | #An | #En | #Cr | #An | #En | #Cr | #An | #En | #Cr | #An | #En | #Cr | #An | #En | #Cr | #An | #En |
| Base | 6 | 6 | 11 | 47 | 54 | 124 | 112 | 122 | 273 | 0 | 0 | 2 | 21 | 27 | 99 | 40 | 47 | 247 |
| Temperature | 29 | 31 | 51 | 95 | 104 | 229 | 137 | 156 | 332 | 7 | 7 | 20 | 50 | 60 | 216 | 63 | 74 | 294 |
| Instruct | 50 | 55 | 68 | 107 | 117 | 190 | 153 | 172 | 302 | 7 | 8 | 13 | 66 | 81 | 167 | **85** | 100 | 273 |
| Token-Blocking | 4 | 4 | 9 | 98 | 109 | 91 | 138 | 162 | 266 | 2 | 2 | 4 | 26 | 32 | 87 | 36 | 47 | 251 |
| TPV $\alpha = 5$ | 4 | 5 | 12 | 37 | 42 | 118 | 109 | 117 | 276 | 0 | 0 | 2 | 22 | 26 | 95 | 35 | 43 | 235 |
| TPV $\alpha = 100$ | 26 | 30 | 41 | 70 | 77 | 170 | 102 | 113 | 273 | 4 | 5 | 7 | 28 | 34 | 118 | 38 | 46 | 223 |
| TPV $\alpha = 5$ Ins | 51 | 55 | 67 | 103 | 112 | 193 | **159** | 175 | 315 | 12 | 12 | 13 | 58 | 72 | 164 | 80 | 95 | 268 |
| TPV $\alpha = 100$ Ins | **91** | 118 | 119 | **124** | 170 | 227 | 142 | 192 | 284 | **27** | 33 | 70 | **80** | 96 | 176 | 81 | 104 | 211 |

associated with overthinking, such as "Wait", "Alternatively", "But", and "Umm", by setting their logits to negative infinity.

Table 2 reveals four notable trends: **(1) The impact of $\alpha$:** Increasing $\alpha$ in our method from 5 to 100, both with and without the instruction-based acceleration used in baseline (iii), generally increases the numbers of completed, ended, and correct answers produced by the model, providing further evidence that our intervention method influences the thinking length. In particular, on Math-500, larger values of $\alpha$ increased the number of correct answers by 42,39,7 on token budget of 512,1024 and 2048 respectively. Similarly, on GSM-8K, the number of ended answers increases by 15, 19, and 5 for token budgets of 256, 512, and 1024, respectively.

**(2) Comparing Acceleration Baselines Against the Base Model:** Baselines (ii) and (iii) accelerate the base model using prompting responses and temperature-based ensembling. Unsurprisingly, in most regimes both methods increase all three metrics, demonstrating that they are strong baselines against which to evaluate our overclocking approach. Specifically, on the Math-500 dataset with a 512-token budget, the number of correct answers rises from 36 (base) to 54 under the temperature baseline and 53 under the instruction-based baseline, while the number of answers increases from 38 to 59 for both methods. Finally, the number of completions increases from 28 for the base model to 43 for the temperature baseline and 52 for the instruction-based baseline. Similarly, under the same 512-token limit on GSM-8K, both acceleration strategies yield improvements. The temperature-based approach increases correct responses from 221 to 237, while the instruction-driven method produces 224 correct answers, and the total answers climb from 223 to 249 with the temperature strategy and to 237 with instruction prompts. Completions also see a marked rise, increasing from 28 for the base model to 43 under the temperature baseline and to 52 with instruction-based acceleration.

**(3) Comparing Our Method Against Baselines:** Although these baselines perform strongly and the temperature-based baseline requires roughly five times more compute, our method outperforms both by producing more correct answers and more decisive responses. In particular, under low compute budgets of 256 or 512 tokens, our approach with $\alpha = 100$ increases the number of correct answers on Math-500 by at least 80% in the 512 token-budget regime and boosts correct responses on GSM-8K by an average of 80% across the 256 and 512 token settings. For example, on Math-500 in the 512 token-budget regime, correct answers rise from 54 under the temperature baseline and 53 under the instruction baseline to 100 with our method, resulting in a gain of more than 46 correct answers, while the number of completed answers increases from 59 under both baselines to 106, a gain of 47. Remarkably, these increases in correct answers do not come at the cost of more errors, as the error rate remains unchanged. This suggests our method shortens reasoning without increasing errors, promoting more decisive thinking. Our findings for compute budgets larger than 512 generally follow the same trend, showing improvements in the number of correct answers in most regimes, without an increase in the error rate.

**(4) Complementary Contribution:** Although our empirical findings approve that our method is more effective than the baselines, there are still cases where our method lags behind the prompting-based approach (denoted as 'Instruct'). One prominent example is the regime of token-budget of 2048 on the Math 500, where the instruct baseline answers correctly 10% more than our method. A similar trend was also observed in Table 3, with experiments conducted using LLaMA instead of the Qwen DeepSeek variants. This observation raises the question of whether the improvements are orthogonal or if the two methods directly compete. To investigate, we conducted dedicated experiments in which

the instruction-based prompting technique was combined with our intervention method and compared against each individual approach. As shown in the last two rows of both tables, this hybrid approach consistently yields the best performance across most regimes, achieving an improvement of $66\%$ on average and $285\%$ at max over the instruction-based baseline, and $223\%$ on average and $1416\%$ at max over the base model. These findings suggest that our method is complementary to prompting strategies and can be effectively integrated with other acceleration techniques.

As a minimal intervention check beyond the math setting, we also applied TPV intervention on GPQA using DEEPSEEK-R1-DISTILL-QWEN-32B. After using the first 30 GPQA Diamond problems (with 5 sampled trajectories each) for training and validating the TPV regressor, we evaluated interventions on the remaining 168 problems. Using the GPQA-trained TPV, we tested $\alpha \in \{5, 10\}$ under two thinking budgets (5k and 10k tokens). As shown in Table 4, a stronger intervention ($\alpha = 10$) reliably increases the number of completed and correct trajectories at identical budgets (e.g., correct: $54 \rightarrow 59$ at 5k, $71 \rightarrow 75$ at 10k).

Table 4: TPV intervention on GPQA with DEEPSEEK-R1-DISTILL-QWEN-32B under two thinking budgets (5k and 10k tokens). We report the number of correct (#Cr), answered (#An) and ended (#En) trajectories.

| Method | 5k | | | 10k | | |
|---|---|---|---|---|---|---|
| | #Cr | #An | #En | #Cr | #An | #En |
| Base | 54 | 70 | 67 | 71 | 97 | 98 |
| TPV $\alpha = 5$ | 56 | 69 | 66 | 70 | 99 | 97 |
| TPV $\alpha = 10$ | 59 | 85 | 83 | 75 | 109 | 110 |

**Efficiency** We conduct a series of intervention experiments on the Math-500 and GSM8K datasets by varying the intervention parameter $\alpha$ to overclock the model's thinking phase. Figure 6 shows that increasing $\alpha$ consistently reduces the length of the thinking phase, making the reasoning process more efficient. These findings support our hypothesis that TPV functions as an active control signal in the model's internal computation, rather than being a passive correlate. In particular, when applying our method with $\alpha = 100$ to the DeepSeek-R1 LLaMA model using the prompting strategy (baseline iii) on the GSM8K dataset, the average number of tokens decreases from approximately 500 to fewer than 350, resulting in a 30% reduction in compute. Moreover, all positive values of $\alpha$ consistently accelerate the thinking phase relative to the baseline ($\alpha = 0$), while also improving its effectiveness, as shown in Tables 2–3. More-

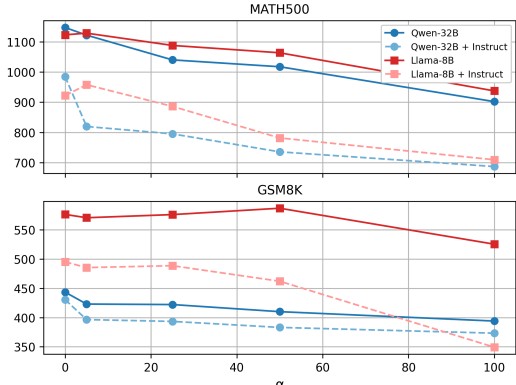

Figure 6: **Impact of $\alpha$ on Intervention:** The x-axis represents different values of $\alpha$ in $\{0, 5, 25, 50, 100\}$, where $\alpha = 0$ indicates no intervention and higher values of $\alpha$ amplify the amount of intervention. The y-axis shows the average number of tokens required to complete the answer. Results are obtained using DeepSeek-R1 models.

over, in Table 8 at Appendix B, we conduct experiments with negative values of $\alpha$, which, as expected, cause down-clocking, namely, lead to longer responses and fewer answered, ended, and correctly answered questions. For additional qualitative examples, see examples in Appendix G.

**Coherence & Fluency** Beyond evaluating text quality and generation efficiency, we now assess the coherence and fluency of the model outputs. To this end, we employ three automated metrics: (i) *Perplexity*, computed with GPT2-Large as the exponential of the average negative log-likelihood, and (ii) *Distinct-1* and *Distinct-2*, which measure the proportion of unique unigrams and bigrams, respectively, among all generated tokens (higher values indicate greater lexical diversity). We report summary statistics (Average, Median, Standard Deviation, Minimum, and Maximum) across three inference settings: the baseline method, denoted as 'Base', and two variants of our intervention method with $\alpha = 5$ and $\alpha = 100$. As shown in Table 7, increasing $\alpha$ results in only minor changes to Perplexity and Distinct-1/2, suggesting that our intervention method preserves the fluency and diversity of the generated text. It is also important to note that at least some of the differences in the metrics arise from differences in sequence length.

**Semantic Analysis of the TPV Direction.** To further investigate whether the TPV encodes superficial lexical cues or instead reflects a deeper latent structure in the model's hidden states, we analyze its semantics by examining its alignment with the unembedding matrix. Specifically, for

Table 5: Top-20 tokens most similar to the TPV.

| Rank | ID | Cosine Similarity | Token |
|---|---|---|---|
| 1 | 220 | 0.037326 |  |
| 2 | 13 | 0.036557 | . |
| 3 | 11 | 0.035818 | , |
| 4 | 12 | 0.034028 | - |
| 5 | 320 | 0.033876 | ( |
| 6 | 25 | 0.032234 | : |
| 7 | 2293 | 0.029799 | — |
| 8 | 7 | 0.029604 | ( |
| 9 | 151643 | 0.029590 | <eos> |
| 10 | 30 | 0.027572 | ? |
| 11 | 62 | 0.026670 | _ |
| 12 | 198 | 0.026608 | \n |
| 13 | 481 | 0.026370 | - |
| 14 | 14 | 0.026264 | / |
| 15 | 26 | 0.025832 | ; |
| 16 | 271 | 0.025007 | \n\n |
| 17 | 6 | 0.024798 | ' |
| 18 | 304 | 0.024312 | in |
| 19 | 323 | 0.024247 | and |
| 20 | 0 | 0.024190 | ! |

Table 6: Cosine similarity between TPV and special/meta tokens.

| Token | ID | Sim. |
|---|---|---|
| *Special Reasoning Tokens* | | |
| `<think>` | 151648 | $-0.013006$ |
| `</think>` | 151649 | $-0.011011$ |
| *Custom Meta Tokens* | | |
| again | 32771 | $-0.006912$ |
| Hmm | 80022 | $-0.011941$ |
| Wait | 14190 | $0.002348$ |
| problem | 34586 | $-0.005753$ |
| so | 704 | $0.011249$ |
| think | 26865 | $-0.003059$ |
| Double-check | 7378 | $-0.005538$ |
| Right | 5979 | $0.000166$ |
| Okay | 32313 | $-0.008491$ |
| But | 3983 | $0.003591$ |
| Hold on | 47427 | $-0.004608$ |

Table 7: **Coherence and fluency evaluation for generated text.** Comparison of *Perplexity*, *Distinct-1*, and *Distinct-2* across different statistics (Average, Median, Standard Deviation, Minimum, and Maximum) for the Baseline (denoted by 'Base'), $\alpha = 5$, and $\alpha = 100$ settings.

| Statistic | Perplexity | | | Distinct-1 | | | Distinct-2 | | |
|---|---|---|---|---|---|---|---|---|---|
| | Base | $\alpha = 5$ | $\alpha = 100$ | Base | $\alpha = 5$ | $\alpha = 100$ | Base | $\alpha = 5$ | $\alpha = 100$ |
| Average | 5.75 | 5.72 | 5.65 | 0.2307 | 0.2268 | 0.2329 | 0.5841 | 0.5713 | 0.5504 |
| Median | 5.56 | 5.52 | 5.40 | 0.2188 | 0.2151 | 0.2144 | 0.5821 | 0.5733 | 0.5601 |
| Std Dev | 1.64 | 1.63 | 2.07 | 0.0565 | 0.0591 | 0.0761 | 0.0768 | 0.0770 | 0.1031 |
| Min | 2.40 | 2.86 | 1.59 | 0.1061 | 0.1094 | 0.0802 | 0.2893 | 0.3558 | 0.1830 |
| Max | 18.83 | 15.47 | 22.01 | 0.4507 | 0.5000 | 0.5718 | 0.8380 | 0.7989 | 0.8261 |

the TPV trained on Deepseek-Distill-R1-Qwen-32B's hidden states over the MATH500 dataset, we compute cosine similarities between its direction and all rows of the model's unembedding matrix, and report the top-20 most similar tokens in Table 5, as well as a curated set of "meta" tokens that commonly occur in reasoning traces in Table 6 (e.g., *"Okay", "Wait", "Hold on", "Right"*). The results reveal three key observations. First, the TPV is not aligned with these meta tokens: their cosine similarities are small and of mixed sign, and in several cases contradict the token-level correlations reported in Appendix E (e.g., "Okay" shows a negative similarity despite correlating positively with progress). This discrepancy indicates that TPV is not relying on these tokens as isolated lexical markers as predictive features. Second, TPV is not aligned with the `</think>` token (negative cosine similarity), ruling out the hypothesis that it approximates the likelihood of terminating the thinking phase. Third, the tokens most similar to TPV are dominated by generic punctuation and formatting symbols (e.g., ".", ",", "-", "("), all with very small cosine values ($\sim$0.03). These tokens are ubiquitous across text and provide only weak structural cues, making them unlikely candidates for driving a systematic progress estimator. Taken together, this analysis suggests that the TPV captures a latent geometric direction that is not reducible to a small dictionary of phrase patterns or token-level heuristics. When combined with our intervention experiments, which show that shifting hidden states along the TPV direction causally accelerates or decelerates reasoning, these findings support the interpretation of TPV as reflecting an internal progress representation rather than a superficial pattern-matching mechanism.

**Qualitative Analysis** We analyze the impact of TPV intervention on different thinking patterns. Fig. 1(b) compares two thinking sequences generated by the DeepSeek-R1-Distill-Qwen-32B model—with and without intervention. The original sequence is marked by hesitation and verbosity, while the TPV-accelerated version is significantly more concise, using nearly six times fewer tokens. Notably, both trajectories arrive at the correct answer. See Appendix G.1 for an additional overthinking samples.

To better understand how progress is monitored and how it is modified during the thinking phase, we explored cases in which the behavior of the indicator is not monotonic, see Appendix D. For example it is often the case that the progress prediction increases, until a noticeable drop occurs. This drop in

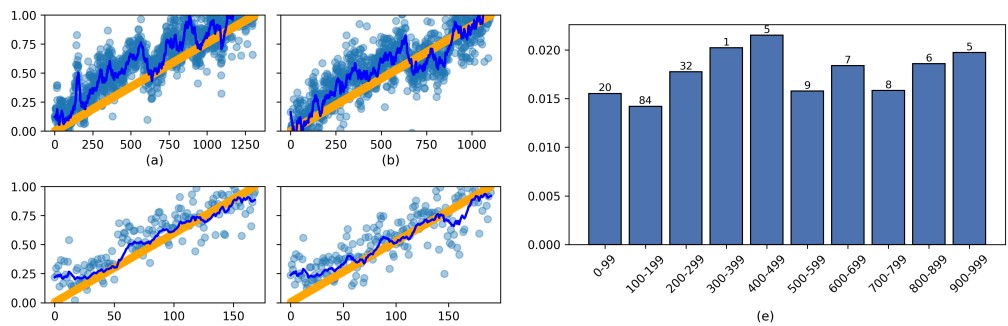

Figure 7: **Robustness of predicted progress under varied prompting styles and reasoning lengths. Left:** Predicted and smoothed progress trajectories for the problem "How many 3-digit numbers can be formed using the digits 1 through 9 (no zeros), with no repeated digits?" under four different prompts: (a) "Please reason step by step." (b) "Think carefully and slowly. Provide a detailed explanation." (c) "Think quickly and provide a concise answer." (d) "Present all necessary calculations or arguments concisely; avoid unnecessary elaboration or verbosity." **Right:** (e) Mean MSE test loss from both Math500 and GSM8K datasets, grouped by thinking phase length (binned). The number of trajectories in each bin is shown above each bar.

predicted progress often coincides with self-verification behavior, where the model attempts to solve the problem again using an alternative approach.

A deeper investigation into the reliability of TPVs in estimating a model's position within its reasoning process is presented in Figure 7. Their performance is evaluated under two additional conditions: (i) varied prompting strategies and (ii) differing reasoning sequence lengths. Figures 7(a–d) demonstrate that TPVs remain effective across diverse instructions, distinct from the original prompt used during training (see caption for the prompts). Figure 7(e) further shows consistently low test loss across bins of different thinking sequence lengths, indicating robustness to varying reasoning depths.

To further explore TPV's capacity to capture and explain different reasoning dynamics, we examine how predicted progress responds to specific tokens that suggest either hesitation ("wait") or advancement ("right"). We quantify this behavior, by measuring the average effect of individual tokens on predicted progress values across the dataset, see Appendix E. The analysis shows that certain words do tend to significantly shift the predicted progress indicator in a characteristic direction.

## 5 CONCLUSIONS

The Nelson and Narens metacognition model (Nelson, 1990) distinguishes two interacting levels: the object level, which performs cognitive operations, and the meta level, which engages in monitoring and control over the object-level processes. We show that these two information flows, monitoring and control, are manifested in LLMs, as described in Sections 3.1, and 3.2, respectively. Similarly, Zimmerman (2002) describe the process of self-regulated learning and highlights monitoring and self-reflection as two essential components. In this context, our work shows that LLMs can estimate the number of reasoning steps required to reach a desired answer, and that this estimation evolves in real time based on the tokens generated.

On the applicative side, our method represents an important step toward making reasoning models more transparent and controllable. At the time of writing, millions of users interact with reasoning models such as GPT 5.0 (OpenAI, 2025b) or DeepSeek-R1. During inference, particularly in the "thinking phase," users are often left without visibility into the model's internal progress or how close it is to producing an answer. By introducing an interactive progress bar that visualizes the model's internal status during reasoning, we make these models more predictable, responsive, and user-friendly. Furthermore, our approach introduces a plug-and-play intervention mechanism that enables real-time acceleration of the reasoning process through overclocking. This feature is especially useful for users who have an intuition about the complexity of their task and wish to adapt the model's reasoning depth accordingly. This contributes to the ongoing effort to make reasoning-capable language models more interactive, adaptive, and aligned with human needs.

## 6    ETHICS STATEMENT

This work investigates methods for monitoring and controlling reasoning path lengths in LLMs. Improving controllability of reasoning can enhance reliability, efficiency, and safety in real-world deployments. Such techniques may also reduce computational overhead and energy consumption. Additionally, similar to other work in mechanistic interpretability, our approach can make LLMs more transparent and easier to collaborate with, further supporting responsible use and alignment with human intentions. We do not anticipate direct negative societal impacts beyond those already associated with the broader use of LLMs.

## 7    REPRODUCIBILITY STATEMENT

We provide the source code for the key experiments and include detailed descriptions of the full configurations for all experiments, along with instructions for setting up and evaluating the models. All datasets, models, prompts, and metrics used in each experiment are explicitly stated in the paper to ensure full transparency and reproducibility.

## 8    THE USE OF LARGE LANGUAGE MODELS (LLMS)

ChatGPT (OpenAI, 2025a) was used as a general-purpose writing and editing assistant in the preparation of this manuscript. Its role was limited to helping polish the presentation: rephrasing sentences for clarity, improving grammar and flow, suggesting alternative wording, and condensing or expanding explanations when asked. Importantly, ChatGPT was not involved in the research ideation, design, implementation, analysis, or interpretation of results.

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

## A   EXPERIMENTAL SETUP

This section describes the experimental setup used for the empirical studies in Sec. 4. All experiments were conducted in PyTorch with official Hugging Face models and DeepSeek prompts. Each run used a single A100 GPU and finished within at most two days.

Table 8: **Negative** $\alpha$. Effect of negative $\alpha$ values on text quality and effectiveness.

| $\alpha$ | #Answered | #Correct | #Incorrect | #Ended | Avg. Tokens |
|---|---|---|---|---|---|
| $-100$ | 31 | 31 | 0 | 26 | 1412 |
| $-5$ | 34 | 34 | 0 | 30 | 1241 |
| 0 | 40 | 39 | 1 | 38 | 1065 |
| 5 | 41 | 40 | 1 | 36 | 1032 |
| 10 | 40 | 39 | 1 | 36 | 1053 |
| 25 | 40 | 39 | 1 | 36 | 1048 |
| 50 | 40 | 39 | 1 | 35 | 874 |
| 100 | 42 | 39 | 3 | 39 | 891 |
| 200 | 27 | 24 | 3 | 28 | 563 |

**Datasets.** We evaluated performance on two publicly available mathematics datasets: GSM-8K and Math-500. For Math-500, we randomly sampled 80 problems to train our loading-bar regressor and used the remaining 420 problems as the test set. For GSM-8K, we randomly sampled 330 problems, reserving the first 30 for training and the remaining 300 for testing. These datasets were chosen because they effectively assess the reasoning capabilities of LLMs, as reasoning has been shown to be remarkably effective on mathematical problems.

## B    EFFECT OF NEGATIVE $\alpha$ VALUES

We further investigate the effect of negative values of $\alpha$, which we expect to promote intervention in the opposite direction of overclocking. Table 8 reports results on 50 problems from the MATH-500 dataset. As expected, negative $\alpha$ consistently increases the average number of generated tokens. For example, with $\alpha = -100$ the response length grows by a factor of 1.32, reflecting more verbose and hesitant reasoning. This behavior also reduces the number of ended, answered, and correctly answered questions. In particular, with $\alpha = -100$ the model answers 9 fewer questions and produces 8 fewer correct responses compared to the baseline.

## C    ROBUSTNESS OF TPV VECTORS MONITORING ACROSS LAYERS

To further demonstrate the robustness of our TPV representations, we evaluate their extraction across different model layers. The results, summarized in Fig. 8, highlight the robustness of the TPV representations across layers and underscore the consistency of our findings. In particular, the train and test regression MSE loss across layers remain relatively low, below 0.06, except in the first layer.

## D    NON-MONOTONIC PROGRESS PREDICTION

We perform a qualitative analysis of TPV-predicted progress trajectories and find that non-monotonic patterns often correspond to self-verification behaviors in the model's reasoning. In such cases, the model reapproaches the problem using a different method to confirm its solution. For instance, as illustrated in Figure 9, the model first solves a combinatorics question using the counting principle, then revalidates its answer by applying the permutations formula. This second solving attempt begins at token 665—indicated by a vertical line—which coincides with a clear drop in the predicted progress, reflecting the model's re-evaluation phase.

## E    EFFECT OF SPECIFIC WORDS ON PREDICTED PROGRESS

When examining thinking sequences qualitatively, certain tokens frequently appear in contexts that suggest either hesitation or progression toward concluding the reasoning process. Words such as "hmm," "wait," and "right" often signal meaningful transitions in the model's internal state. Some of these tokens reflect a recognition of inconsistency or error, prompting the model to adjust its

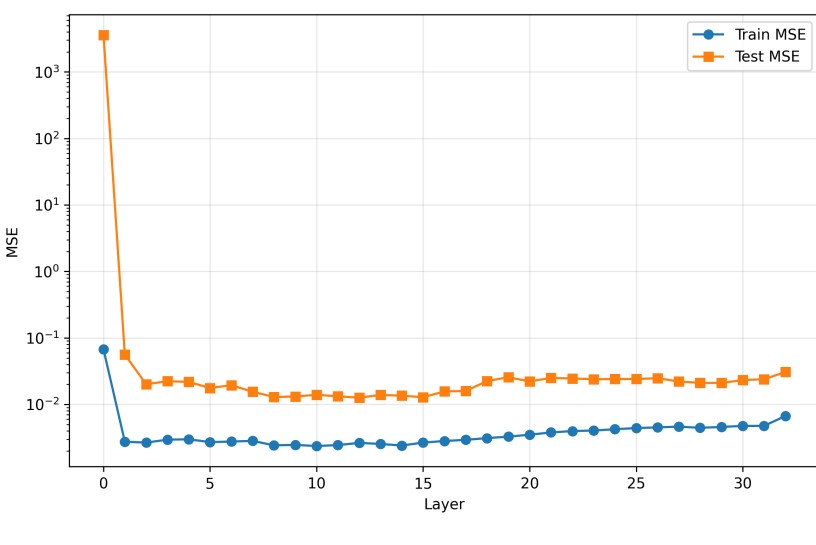

Figure 8: **TPV regression across layers.** A TPV regressor is trained independently on each layer of the model, and the train and test MSE values are reported. Lower is better.

(Q) How many 3-digit numbers can be formed using the digits 1 through 9 (no zeros), with no repeated digits?

(T 1) Okay, so I have this problem: How many 3-digit numbers ...
(T 42) First, I know that ...
(T 201) Now, moving on to the second digit ...
(T 332) Now, according to the counting principle ...
(T 572) It is indeed 504. Okay, that seems right ...
(T 630) Let me think about another way ...
**(T 665) The formula for permutations is ...**
(T 883) I think that solidifies the answer ...

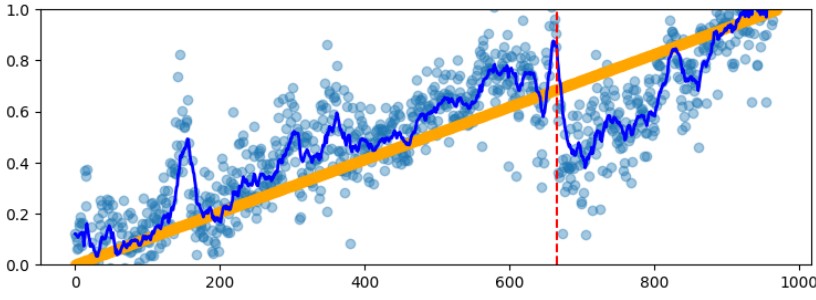

Figure 9: A math question and the associated thinking sequence generated by DeepSeek-R1-Distill-Qwen-32B, with each row indicating the index of its first token. The sequence is paired with its TPV-predicted progress trajectory, highlighting the effect of self-verification behavior within the model's reasoning process. A vertical line at token index 665 marks a notable drop in predicted progress, aligning with the point where the model begins re-solving the problem.

reasoning strategy. Others indicate the completion of a reasoning step and a shift toward the final solution. TPVs enable a principled way to quantify this behavior.

We analyze the average influence of individual tokens on predicted progress values across the dataset. As illustrated in Figure 10, tokens with the most negative impact—such as "again," "hmm," "wait," and "problem"—are closely tied to hesitation or reevaluation. In contrast, tokens like "right," "so,"

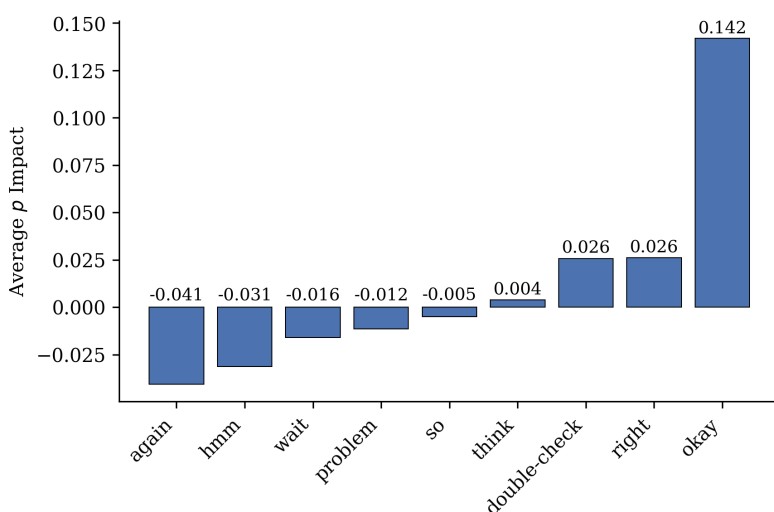

Figure 10: Average change in $\bar{p}$ values caused by significant tokens across the dataset, highlighting their impact on predicted progress.

and "okay" are associated with upward shifts in predicted progress, signaling growing confidence and momentum in the model's reasoning process.

# F  ADDITIONAL DETAILS: TPV NORMS

For completeness, we report here the $\ell_2$ norms of the TPV directions learned for each dataset and each model.

Table 9: TPV vector norms for DEEPSEEK-R1-DISTILL-QWEN-32B (Q32B) and DEEPSEEK-R1-DISTILL-LLAMA-8B (L8B).

| Dataset | Q32B Norm | L8B Norm |
|---|---|---|
| Math500 | 0.256 | 0.120 |
| GSM8K | 0.250 | 0.137 |
| HumanEval | 0.243 | 0.116 |
| StrategyQA | 0.159 | 0.117 |
| GPQA | 0.371 | 0.126 |

These norms tend to be stable across datasets and models, showing moderate variation that reflects differences in problem domains.

# G  ADDITIONAL QUALITATIVE ANALYSIS

To highlight the impact of our intervention in mitigating overthinking, we present a qualitative analysis in Examples 1–3. Examples 2–3 demonstrate that while both the baseline model and its over-clocked variant arrive at the same answer, the over-clocked model does so more decisively and with significantly fewer thinking tokens. In contrast, Example 1 shows a case where the base model does not succeed in providing an answer within the token limit, in contrast to the over-clocked variant, which provides an answer after 259 thinking tokens.

## G.1  OVERTHINKING EXAMPLE

Example 4 illustrates a mathematical problem where the original model, when prompted, delves into a loop of self-verification and indecisiveness, raising repeating arguments and calculations without

**Prompt**

Suzanne walks four miles every third day. What is the fewest number of miles she can walk in February? Please reason step by step, and put your final answer within \boxed{}

**TPV $\alpha = 100$ (259 Thinking Tokens)**

<think> Suzanne walks 4 miles every 3 days. To find the fewest number of miles she can walk in February, we need to determine how many 3-day periods are in 28 days (a non-leap year February).

1. **Calculate the number of 3-day periods in 28 days:**

$$\frac{28}{3} \approx 9.33$$

Since Suzanne can't walk a fraction of a 3-day period, we take the integer part, which is 9.

2. **Determine the total miles walked:**

$$9 \times 4 = 36 \text{ miles}$$

3. **Check if 28 days is a multiple of 3:**

$$28 \div 3 = 9 \text{ with a remainder of } 1$$

Since 28 is not a multiple of 3, 9 periods of 3 days account for 27 days, leaving 1 day. Suzanne doesn't walk on the 28th day, so 36 miles is the minimum.
**Final Answer** The number of ways is \boxed{36}. </think>

**Original (Continues past 2048 tokens)**

<think> Okay, so I need to figure out the fewest number of miles . . .
First, I should know how many days are in February. . . . maybe I can assume it's a non-leap year with 28 days? Or does it matter? Let me think. . . .
Let me list the days she walks:
Day 1, 4, 7, 10, 13, 16, 19, 22, 25, 28.
**Wait**, that's 10 walks. . . . Maybe she can start on a later day to minimize the number of walks in February. . . .
But how late can she start? . . . if she starts on day 26, for example, she would walk on day 26, then day 29, but February only has 28 days, . . . she would only walk once in February. . . .
**Wait**, actually, the problem says she walks four miles every third day. . . .
**But wait**, if she starts on day 26, does that mean she didn't walk on day 23? . . . she should have walked on day 23 as well. . . .
If she walks every third day, that means her walking days are spaced three days apart. So, if she walks on day x, she must have walked on day x - 3, x - 6, etc., and will walk on day x + 3, x + 6, etc. . . .
**Wait**, so starting on day 2 or 3, she has 9 walks, whereas starting on day 1, she has 10 walks.
**But earlier**, I thought she could start on day 26 and only walk once, but that seems incorrect because her walking schedule is every third day, so she can't just start on day 26 without having walked on day 23, 20, etc. . . .

Example 1: An example of overthinking through first presenting an argument not aligned with the constraints of the problem and the effort of reconciling it (right); and a straightforward answer presenting a simple solution (left).

converging to a conclusion within the limit of 2048 tokens. The accelerated version reaches the correct answer while terminating in less than half of the token limit.

**Prompt**

Compute the 9th Fibonacci number, assuming F(1) = 1. Please reason step by step, and put your final answer within \boxed{}

**TPV $\alpha = 100$ (293 Thinking Tokens)**

<think> To compute the 9th Fibonacci number, I start by understanding the Fibonacci sequence. The sequence is defined such that each number is the sum of the two preceding ones, with the first two numbers being 1.
I will list out the Fibonacci numbers up to the 9th term. Starting with F(1) = 1 and F(2) = 1, I can calculate each subsequent term by adding the two previous terms:
F(3) = F(2) + F(1) = 1 + 1 = 2
. . .
F(9) = F(8) + F(7) = 21 + 13 = 34
By following this sequence, I find that the 9th Fibonacci number is 34. </think>

**Original (1872 Thinking Tokens)**

<think> Okay, so I need to find the 9th Fibonacci number, and they mentioned that F(1) is 1. Hmm, Fibonacci numbers. . .
I remember they start with 0 and 1, but . . .
The Fibonacci sequence is where each number is the sum of the two preceding ones. So, if F(1) is 1, then what is F(2)? I think it's also 1 because usually, the sequence starts with 0, 1, 1, 2, 3, 5, etc. . . .
Let me write down the sequence step by step to make sure I get it right.
F(1) = 1
F(2) = 1
. . .
F(9) = F(8) + F(7) = 21 + 13 = 34
**Wait, so F(9) is 34?**
**Let me double-check** my calculations to make sure I didn't make a mistake.
Starting from F(1) = 1:
1. F(1) = 1
2. F(2) = 1
. . .
9. F(9) = 21 + 13 = 34
Yes, that seems correct. Each time, I'm adding the two previous numbers to get the next one. So, the 9th Fibonacci number is 34. . . .
**let me recall another way** . . .
I remember there's Binet's formula, which uses the golden ratio. The formula is:

$$F(n) = (\Phi^n - \Psi^n)/\sqrt{5} \ldots$$

Let me try using this formula to compute F(9)
. . .
First, compute . . .
Now, compute . . .
Let me compute this step by step: . . .
. . .
Similarly, compute . . .
. . .
Now, plug these into Binet's formula:

$$F(9) = (\Phi^9 - \Psi^9)/\sqrt{5} \approx$$

$(76.013-(-0.01316))/2.23607 \approx \ldots \approx 34.0$

Wow, that's exactly 34. So, using Binet's formula also gives me 34 for F(9). That confirms my earlier result.
**Alternatively**, I could have used the recursive formula . . .
**Just to make sure**, let me list out the Fibonacci numbers from F(1) to F(9):
1. F(1) = 1
2. F(2) = 1
. . .
9. F(9) = 34
Yep, that's consistent. So, I'm confident that the 9th Fibonacci number is 34. </think>

Example 2: An example of overthinking through excessive self-verification and redundant, duplicated computation in the thinking process of the base model (right), which is mitigated in the text produced by our model (bottom left).

**Prompt**

Solve X + Y = 5 , 3X - Y = 8. Please reason step by step, and put your final answer within `\boxed{}`

**TPV $\alpha = 100$ (202 Thinking Tokens)**

<think> To solve the system of equations: 1. Add the two equations to eliminate Y:

$$(X + Y) + (3X - Y) = 5 + 8$$

$$4X = 13$$

$$X = \frac{13}{4}$$

2. Substitute X back into the first equation to find Y:

$$\frac{13}{4} + Y = 5$$

$$Y = 5 - \frac{13}{4} = \frac{20}{4} - \frac{13}{4} = \frac{7}{4}$$

3. Final Answer:

$$X = \frac{13}{4}, \quad Y = \frac{7}{4}$$

</think>

**Original (892 Thinking Tokens)**

<think> Okay, so I have these two equations here: X + Y = 5 and 3X - Y = 8. I need to solve for X and Y.
Hmm, let me think about how to approach this. I remember from class that there are a couple of methods to solve systems of equations, like substitution and elimination. Maybe I can use one of those here.
Let me write down the equations again . . .
Hmm, . . . Let me try that.
Adding equation 1 and equation 2:
(X + Y) + (3X - Y) = 5 + 8
Simplify the left side:
X + Y + 3X - Y
Combine like terms:
X + 3X is 4X, and Y - Y is 0. So, 4X = 13
Wait, 5 + 8 is 13, so 4X = 13. Then, to solve for X, I can divide both sides by 4.
X = 13/4
**Hmm**, . . .
Okay, now that I have X, I can plug it back into one of the original equations to find Y.
Let me use equation 1 because it looks simpler: X + Y = 5.
Substituting X = 13/4 into equation 1:
13/4 + Y = 5
To solve for Y, I can subtract 13/4 from both sides.
Y = 5 - 13/4
**Hmm**, 5 is the same as 20/4, right? So, . . .
So, Y = 7/4, which is 1.75 or 1 and 3/4.
**Let me double-check** my work . . .
Starting with the two equations: . . .
Which simplifies to . . .
Then, plugging X back into equation 1: . . .
Subtract 13/4: . . .
That also seems correct.
**Just to be thorough**, let me plug . . .
. . .
Yes, that works out. So, both equations are satisfied with X = 13/4 and Y = 7/4.
I think that's it. I used the elimination method by adding the two equations to eliminate Y, solved for X, then substituted back to find Y. Checked my solution in both equations, and it works.
So, I feel confident that X is 13/4 and Y is 7/4.
</think>

Example 3: An example of overthinking through excessive self-verification and redundant, duplicated computation in the thinking process of the base model (right), which is mitigated in the text produced by our model (bottom left).

1026
1027
1028
1029
1030
1031
1032
1033
1034
1035
1036
1037
1038
1039
1040
1041
1042
1043
1044
1045
1046
1047
1048
1049
1050
1051
1052
1053
1054
1055
1056
1057
1058
1059
1060
1061
1062
1063
1064
1065
1066

**Prompt**

In how many ways can 8 people sit around a round table if 3 of the people – Pierre, Rosa, and Thomas – all want to sit together? (Two seatings are considered the same if one is a rotation of the other.) Please reason step by step, and put your final answer within \boxed{}

**TPV $\alpha = 100$ (806 Thinking Tokens)**

<think> Okay, so I have this problem: 8 people want to sit around a round table, . . . Alright, let's break this down. I remember that for circular permutations, the number of ways to arrange n people is (n-1)! . . . I think the method is correct because we're treating the block as a single unit, so the number of units is 6, and the circular permutations of 6 units is 5!. Then, within the block, 3! arrangements. So, 5! * 3! = 720. I think that's solid. I don't see any mistakes in the reasoning. So, the answer should be 720. **Final Answer** The number of ways is \boxed{720}. </think>

**Original (Continues Past 2048 Tokens)**

<think> Okay, so I have this problem where I need to figure out how many ways 8 people can sit around a round table. The catch is that Pierre, Rosa, and Thomas all want to sit together. Hmm, circular permutations can be a bit tricky, but let me think through this step by step. . . . So, if I consider Pierre, Rosa, and Thomas as one block **But wait**, . . .
**But hold on**, I want to make sure I didn't miss anything. . . .
**Is there another way to think about this?** Maybe by
**Wait**, no, that might complicate . . .
**Wait**, maybe I made a mistake. . .
**Wait**, perhaps the first approach is wrong . . .
**Wait**, no, I think I confused. . .
**Wait**, maybe I need to think differently. . . .
**Wait**, so if I fix Pierre's seat, then the block is fixed relative to Pierre, . . .
**Wait**, maybe both approaches are correct, but they are counting different things. . . .
**Wait**, no, . . .**Wait**, let me think again. If we fix Pierre's seat . . .

Example 4: Qualitative examples of indecisive loops and overthinking in the thinking process of the base model (right), which is then mitigated in the text produced by our model (bottom left).
