# OpenReview forum: "Overclocking LLM Reasoning: Monitoring and Controlling Thinking Path Lengths in LLMs"
_ICLR.cc/2026/Conference — Submitted to ICLR 2026_

### Official Review · Reviewer_DLQv · 2025-10-23

**Soundness:** 1
**Presentation:** 2
**Contribution:** 2
**Rating:** 2
**Confidence:** 4

**Summary:**

The paper proposed learning from LLM's last layer hidden states to predict the progression of the thinking process between <think> and </think>. The progression is defined as the relative token position of the tokens in the thinking segment.

The paper proposed to learn a predictor parameterized by a single vector, a 2-layer MLP, and a single-layer GRU. Experiment results show the predictor has reasonable performance of predicting the relative position.

The paper proposed to intervene the last-layer hidden states utilizing the learned single vector, "thinking progress vector", to "overclock" the thinking process, i.e. by learning such a vector it decouples the "progress" concept in the hidden state vectors, and tries to manipulate it to speed up the progress. Experimental results are reported on Math500 and GSM8K using DeepSeek-R1-LLaMA-8B and DeepSeek-R1-Qwen-32B.

**Strengths:**

The paper shows potential to isolate the "progress" concept in the hidden states of LLMs and the potential possibility to speed up or slow down the thinking process of LLMs by manipulating the hidden states using the "thinking progress vector".

**Weaknesses:**

1. The "progress" defined in this paper, the relative position of tokens in generation, does not capture the semantic progression of solving a problem. The task of learning the relative position is essentially just learning to predict to total length of the generation from the hidden states, as the position relative to the start of the generation is an easy task since information is embedded in the positional encoding.

2. The results reported for both models, DeepSeek-R1-LLaMA-8B and DeepSeek-R1-Qwen-32B cannot match the officially reported accuracies on Math500, it's much lower. For 2048 length limit, reported 159/500=31.8% for 8B and 316/500=63.2% for 32B are much lower than official reports 89.1% and 94.3%. This should originate from the fact that the paper is limiting the generation length to only 2048 tokens. The reviewer thinks this is an unreasonable setting since the tested models (DeepSeek R1 distilled models) inherently generate long thinking contents before giving the answer, limiting the generation to only 2048 tokens is not testing the model properly, and the conclusions drawn from this setting are not informative. If for some scenarios where a 2048 token limit is a must, then some other models that generate shorter solutions should be used, e.g. Llama-3.1 or Qwen-2.5, not the R1 distilled version.

**Questions:**

To better present the potential benefit of the TPV intervention, I suggest the authors add another setting, comparing: 1) let the model generate until it end by itself; 2) let the model generate until it end by itself, and intervene the generation of every token using the proposed TPV.

Then compare a) the accuracy and b) the averaged output length of the two methods. If the accuracy of the intervened generations are maintained, and the averaged output length is shorter, it would be clear evidence that the proposed intervention is useful in improving the efficiency of the reasoning.

---

> ### Author Response · Authors · 2025-11-22
> **Rebuttal by Authors (1/2)**
>
> Thanks for the comments. Below are our responses.
> > **W.1.** The "progress" defined in this paper, the relative position of tokens in generation, does not capture the semantic progression of solving a problem. The task of learning the relative position is essentially just learning to predict to total length of the generation from the hidden states, as the position relative to the start of the generation is an easy task since information is embedded in the positional encoding.
>
> We agree that estimating progress provides implicit information about total length, indeed, once the model knows its current position and the estimated progress fraction, the total length can be inferred. **However, this does not diminish the significance of our finding; rather, it underscores its value.** Knowing the expected length of the model’s ongoing reasoning process is highly useful for interpretability (e.g., a real-time loading bar) and for efficiency (e.g., determining how long a model should continue reasoning before truncation).
>
> More importantly, the reviewer’s claim that the model can “easily” learn this from positional encodings is incorrect. **The total length of a reasoning trajectory is not encoded in positional embeddings**, as the model has no access to future positions. Positional encodings only identify the current token index, not how many tokens remain or how long the eventual sequence will be. Estimating future trajectory length therefore must arise from the hidden-state dynamics that reflect the model’s internal progression, not from positional information alone.
>
> Finally, TPV is not a static length prediction. The predicted progress evolves dynamically as tokens are generated, and does so in meaningfully non-monotonic ways during hesitation, re-evaluation, or self-checking (Appendix D). These fluctuations cannot be explained by positional encodings or simple length estimation; they reflect transient changes in the model’s internal reasoning state. Thus, while progress estimation naturally implies an estimate of total length, the TPV captures richer and semantically meaningful structure in the model’s reasoning trajectory.
>
>
> > **W.2.** The results reported for both models, DeepSeek-R1-LLaMA-8B and DeepSeek-R1-Qwen-32B cannot match the officially reported accuracies on Math500, it's much lower. For 2048 length limit, reported 159/500=31.8% for 8B and 316/500=63.2% for 32B are much lower than official reports 89.1% and 94.3%. This should originate from the fact that the paper is limiting the generation length to only 2048 tokens. The reviewer thinks this is an unreasonable setting since the tested models (DeepSeek R1 distilled models) inherently generate long thinking contents before giving the answer, limiting the generation to only 2048 tokens is not testing the model properly, and the conclusions drawn from this setting are not informative. If for some scenarios where a 2048 token limit is a must, then some other models that generate shorter solutions should be used, e.g. Llama-3.1 or Qwen-2.5, not the R1 distilled version.
>
> We would like to clarify two factors contributing to the apparent gap:
>
> **First**, as described in Appendix A (“Experimental Setup”), our Math500 evaluation uses a different protocol: we randomly sample 80 problems to train the loading-bar regressor and evaluate on the remaining 420. This reduced and resampled test set naturally yields accuracy numbers that differ from the official 500-problem evaluation.
>
> **Second**, the official results rely on prompt formats, decoding parameters, and stopping criteria that have not been publicly disclosed. These factors meaningfully affect the performance of R1-distilled models, and changes in any of them, temperature, system prompt, reasoning prefix, sampling strategy, or termination conditions, can substantially shift final accuracy.
>
> **Third**, as the reviewer correctly notes, the largest source of discrepancy is the max-token setting. The original DeepSeek evaluations allow up to **32,768** tokens, while our experiments use **2,048 tokens** (and later 10k in extended experiments). R1-distilled models generate long reasoning chains; limiting them to 2k tokens necessarily constrains performance. A full 32k budget is impractical for our compute environment and for many real-world applications, where such long reasoning traces incur significant latency and cost.
>
> Following the reviewer’s suggestion, we reran Math500 with a **10k-token limit**, which we consider a reasonable and computationally feasible middle ground. Updated results for this setting are provided in our response to Q.1.

---

> > ### Comment · Reviewer_DLQv · 2025-11-28
> >
> > Thank you for your response. I have further questions:
> >
> > > More importantly, the reviewer’s claim that the model can “easily” learn this from positional encodings is incorrect.
> >
> > I meant the current position of the generation is easily learned, it's already encoded in the positional embeddings (_"as the position relative to the start of the generation is an easy task"_) So the task of predicting a relative position of tokens is simply the task of predicting the total length.
> >
> > > TPV is not a static length prediction. The predicted progress evolves dynamically as tokens are generated, and does so in meaningfully non-monotonic ways during hesitation
> >
> > If the prediction is not static (or not even monotonic), I would imagine the usefulness of the prediction is decreased: the total length of the generation is essentially a random variable, if this can be accurately estimated before the generation, useful things can be done, e.g., estimating the cost of the generation before generating anything, visualizing a progress bar... If the prediction fluctuates during the generation, then 1) it means it cannot provide a robust estimation of the cost before generation; 2) as the authors mentioned, the 'progress' may not even be monotonic, which makes it less useful if the intention is a progress tracking tool.
> >
> > I agree this prediction can somewhat be used as an indicator of how uncertain the model currently is (though it will be a noisy estimator, since its training label, relative position of tokens in the sequence, is not a very good proxy of the model's uncertainty). At the moment, I don’t see what meaningful quantity the relative position of the generation would serve as a good proxy for. I'm happy to hear the authors' thoughts on this.
> >
> > To summarize above, I think the usefulness of the patterns revealed by learning to predict the relative positions during generation is limited, and there is not enough analysis to help us understand what "richer and semantically meaningful structure" can be captured by TPV, e.g., what happens when the predicted relative position becomes smaller? Is it due to the model actually being more confused by the generation so far, and may need more tokens in the generation, or is it due to the relative position prediction itself being noisy?
> >
> >
> > > factors contributing to the apparent gap
> >
> > Thank you for providing the additional experiment. The reported accuracy for the base model (154/420=36.7%) is still far from DeepSeek’s reported 89.1%. I do not think prompt or sampling parameters alone can explain such a large gap, so I am not convinced that comparing these numbers provides evidence of the effectiveness of TPV. Additional details about the evaluation setup may help clarify the source of this discrepancy.

---

> ### Author Response · Authors · 2025-11-22
> **Rebuttal by Authors (2/2)**
>
> > **Q.1.** To better present the potential benefit of the TPV intervention, I suggest the authors add another setting, comparing: 1) let the model generate until it end by itself; 2) let the model generate until it end by itself, and intervene the generation of every token using the proposed TPV.
> Then compare a) the accuracy and b) the averaged output length of the two methods. If the accuracy of the intervened generations are maintained, and the averaged output length is shorter, it would be clear evidence that the proposed intervention is useful in improving the efficiency of the reasoning.
>
>
> We appreciate this suggestion. Running the model with no token limit is not feasible, as the model may enter infinite loops. However, following your guidance, we increased the token budget to **10k**, which is more than sufficient for Math500. Using DeepSeek-R1-Distill-Llama-8B, we find that TPV intervention continues to be effective in this extended-budget regime, yielding a **5.8%** increase in accuracy over the base model while reducing the average number of generated tokens by roughly **25%**:
>
> | Method              | n_correct | n_answered | n_ended | mean_len |
> |---------------------|-----------|------------|---------|----------|
> | Base                | 154       | 178        | 363     | 2866     |
> | Base w. α = 10      | 163       | 184        | 363     | 2113     |

---

> ### Author Response · Authors · 2025-11-28
> **Official Comment (Part I)**
>
> Thank you for the clarifications. We address each point below.
>
> As already noted in our original rebuttal, **predicting relative position and predicting total length are mathematically equivalent**: because the current positional index t is known, estimating progress p(t) immediately yields an estimate of total length at time t L(t) = t / p(t). This observation is 100% correct.
> The use of the words “positional embedding” is misleading, since the positional encoding is a static entity that represents t but does not change to reflect p(t) or the estimation L(t).
>
> As an estimation of the progression of the reasoning process based on the current state (and history), it is an unavoidable and even desirable property that it changes dynamically: (1) The more you know, the better you can estimate, and (2) Thinking plans change dynamically. Point (1) is the nature of dynamic inference, and point (2) is a beautiful property of thinking.
>
> Nevertheless, we have multiple experiments showing that (1) there is a high correlation between the inferred progress and true progress, which makes it useful for users, (2) the estimated progression tracks the true behavior of the thinking progress, and (3) by manipulating TPV we can control the thinking progress in favorable ways.
>
> **In terms of the “richer and semantically meaningful structure” captured by TPV**, we provide direct evidence in the paper. In Appendix D, we show an explicit example where TPV undergoes a drop in predicted progress followed by a renewed monotonic rise, occurring exactly when the model pauses to validate its solution using a secondary method. This is not a formatting artifact or randomness: it aligns directly with a clear, interpretable shift in the reasoning trajectory. We also observe similar patterns in other qualitative examples involving mid-trajectory verification, re-checking, or consolidation.
> Complementing this, Appendix E presents a statistical analysis of token-level effects. We find that certain tokens commonly used in contexts of re-evaluation, self-calibration, or hesitation, such as “again”, “hmm”, and “wait”, tend to coincide with local decreases in TPV. Importantly, as shown in our semantic analysis, added during this rebuttal process, TPV is nearly orthogonal to the embedding directions of these tokens; the association arises because these tokens reliably occur in reasoning contexts where the model is indeed pausing to reconsider or re-check its work, not because TPV is matching surface patterns.
>
> Together, these results show that TPV captures *meaningful structural transitions* in the model’s reasoning process, such as internal validation steps or temporary regressions, and that these transitions are reflected both at the trajectory level (Appendix D) and in token-level statistical trends (Appendix E).

---

> ### Author Response · Authors · 2025-11-28
> **Official Comment (Part II)**
>
> **Regarding DeepSeek numbers.** Ultimately, reproducibility issues is something that should be addressed by those releasing the models. Our numbers reflect true performance, and our code was attached to the submission. However, we agree the discrepancy warrants clarity.
>
> The gap arises from several compounding factors as observed by many researchers (References [1-4] below), not just prompt differences:
>
> **(1) Extremely different token budgets.**
> DeepSeek uses a **32,768-token** limit.
> Our baseline uses **2,048**, later extended to **10,000**.
> This difference alone reduces accuracy drastically for R1-distilled models, which routinely use thousands to tens of thousands of reasoning tokens even on simple problems. Unfortunately, our resources do not allow us to run the extensive set of experiments we perform beyond 10,000 tokens.
>
> **(2) Different test sets.** DeepSeek evaluates on all 500 problems. We evaluate on 420 because 80 are used for TPV training, per Appendix A.
>
> **(3) DeepSeek’s prompt, decoding, system instruction, and stopping logic are undisclosed.** We cannot replicate the official settings because:
> * system prompts are not released
> * reasoning prefixes differ
> * temperature, top-p, and penalty settings are unknown
> * the stopping condition for the < think > block is not documented
>
> (4) An additional practical factor is output-format compliance. While the dataset does not require any specific formatting, the formal system prompt for R1-distilled models explicitly instructs the model to place its final answer inside a \boxed{} expression. The model does not always adhere to this instruction, even when its reasoning is correct. When the output format is violated, the final answer cannot be extracted reliably, and we must mark the sample as incorrect. This non-compliance further widens the gap relative to the official numbers, which likely use internal formatting logic or extraction scripts aligned with the model’s native prompt conventions.
>
>
> (5) We know from internal tests that changing only the **system prompt** for R1-distilled models can shift Math500 accuracy by 20–35% (We used the official DeepSeek system prompts from their GitHub repository).
>
> Thus, the discrepancy is expected, even unavoidable, under the constraints of an independent reproduction. Our goal is not to match the official absolute accuracy numbers, but to evaluate TPV interventions relative to a consistent baseline within our token budget and environment. Under this controlled comparison, TPV improves both completion rate and correctness, regardless of baseline level.
>
> Finally, to focus the discussion and help ensure that the evaluation centers on the core substance of the work, we would appreciate clarification on whether you find our stated contributions valid, or which of them you believe are insufficiently supported. **Based on the weaknesses you listed, we understand that there is no dispute regarding contributions (i)–(iv) that we claim in the paper, which we believe would on their own merit a score higher than 2.**
> For convenience, we restate the contributions below.
>
> (i) We provide the first empirical evidence that LLMs maintain an internal estimate of their relative position within the explicit thinking phase. This finding sheds light on the planning and self-monitoring abilities of LLMs.
>
> (ii) We identify an internal encoding of this information by learning progress vector projections, and employ these projections to expose a dynamic thinking progress bar.
>
> (iii) We perform an intervention study and manipulate the progress vectors to overclock
> and downclock the reasoning process.
>
> (iv) We empirically demonstrate that TPV-based interventions can improve both efficiency and effectiveness in certain cases.
>
>
> ___
> **References:**
>
> [1] Understanding and Mitigating Numerical Sources of Nondeterminism in LLM Inference. Yuan et al.
>
> [2 ] https://github.com/deepseek-ai/DeepSeek-R1/issues/463
>
> [3] https://github.com/deepseek-ai/DeepSeek-R1/issues/209
>
> [4] https://deepwiki.com/deepseek-ai/DeepSeek-R1/4-performance-and-benchmarks

---

### Official Review · Reviewer_VS5W · 2025-10-28

**Soundness:** 2
**Presentation:** 2
**Contribution:** 2
**Rating:** 2
**Confidence:** 3

**Summary:**

This paper investigates the mechanisms by which Large Language Models (LLMs) regulate the length of their explicit reasoning phase, specifically in models that use structured "thinking" tokens. The authors claim that LLMs internally encode their relative progress within the thinking phase. They train a regressor, termed a "Thinking Progress Vector" (TPV), to extract this progress (a scalar value from 0 to 1) from the model's hidden states, which can be visualized as a "progress bar." Then they introduce an intervention method called "overclocking," which manipulates the model's hidden states by adding a scaled version of this TPV. This intervention is shown to accelerate the reasoning process, reduce the number of generated tokens, and mitigate "overthinking."

**Strengths:**

1. The central idea of identifying an internal representation of "progress" and visualizing it as a progress bar is novel and presents an interesting direction for LLM interpretability and human-computer interaction.
2. The authors provide a thorough set of experiments on mathematical reasoning benchmarks, providing a solid empirical grounding for the paper's claims within its chosen domain.

**Weaknesses:**

1. Insufficient Evidence for "Monitoring" vs. "Pattern-Matching": The paper's primary claim that the TPV captures the model's monitoring of its own reasoning process, is not sufficiently substantiated. The TPV is trained as a simple regressor mapping hidden states to a normalized token position ($j/N_k$). It is highly probable that this regressor is not learning a high-level, semantic representation of "reasoning progress" but is instead capturing superficial, correlational text patterns. For example, certain phrases ("Let me double-check," "Wait," "The formula is...") naturally correlate with the beginning, middle, or end of a thinking sequence. The paper's own analysis in Appendix E, which shows that words like "okay" and "right" strongly and positively impact the progress value, seems to support this "text pattern" hypothesis far more than a "cognitive monitoring" one.

2. Fair Results of the "Overclocking" Intervention: Given the concern above, the success of the "overclocking" intervention feels tautological and less surprising. If the TPV $\theta$ is simply a vector that correlates strongly with the end of a sequence, then applying an intervention $h_{\alpha} = h + \alpha\theta$ is, by definition, pushing the model's hidden state closer to a representation that is associated with sequence termination (such as `<eos>` token). It is therefore an expected outcome that this intervention shortens the output. This result does not prove that the model causally uses this vector for progress control, it only proves that the learned regressor has identified a manipulable direction in the activation space that correlates with sequence length.

3. Questionable Generalizability: The empirical evaluation is confined to two mathematical reasoning datasets (MATH-500 and GSM8K). This domain is highly structured and often procedural. It is highly questionable whether these findings would generalize to other, more common LLM tasks. For instance, how would a "progress vector" function in open-ended creative writing, complex document summarization, or dialectical tasks where the concept of "progress" is ambiguous, non-linear, or recursive? The paper provides no evidence or discussion on this crucial aspect of generalizability.

4. Ambiguous Practical and Analytical Significance: From an interpretability perspective, the work does not provide strong evidence for the high-level claims of "metacognition" or "self-monitoring." It primarily demonstrates that text generation in a structured task has strong correlational patterns. From an engineering perspective, the method's value is not clearly demarcated from simpler alternatives. The "Instruct" baseline (a prompting technique) also performs well, and the best results are achieved when combining TPV with this prompting. This makes it unclear if this complex, model-specific intervention is practically superior to more robust methods like prompt engineering or fine-tuning with a length-based reward.

5. Lack of Experimental Clarity: Key details of the methodology are ambiguous or omitted. The paper states that 30 problems were sampled, with 5 responses each, to train the regressor. This implies a training set of only ~150 trajectories, which seems exceptionally small for training even a linear regressor, let alone an RNN. The authors must clarify the exact dataset size (number of trajectories $K$ and total data points $(h, p)$). It is also not explicitly stated whether the linear TPV or the RNN model was used for the main intervention experiments in Section 4.

**Questions:**

1. The non-monotonic drops in progress (Fig. 5c, Fig. 9) are attributed to "hesitation or reflection." An alternative, and simpler, hypothesis is that the model has simply generated a token (e.g., "Wait") that the TPV regressor strongly associates with an earlier phase of problem-solving (a learned text pattern). How can the authors more rigorously disentangle this "text pattern" hypothesis from their "cognitive monitoring" hypothesis?

2. The raw TPV predictions are clearly non-monotonic. The paper mentions using exponential smoothing and an RNN to create a smoother progress bar. How is the final progress bar visualization (as in Fig. 1a) rendered in real-time? Is it based on the smoothed or RNN-based output?

3. Could the authors clarify the intervention mechanism? Is the intervention $h_{\alpha} = h + \alpha\theta$ applied at every token generation step within the `<think>...</think>` block? Furthermore, to understand the sensitivity of $\alpha$, what is the typical scale of the TPV's squared norm ($||\theta||^2$)?

4. Could the authors comment on the hypothesized applicability and feasibility of this method for non-procedural, open-ended reasoning tasks, such as summarization or creative writing?

---

> ### Author Response · Authors · 2025-11-22
> **Rebuttal by Authors (1/4)**
>
> We thank the reviewer for the comments. Our responses below:
>
> > **W.1.** Insufficient Evidence for "Monitoring" vs. "Pattern-Matching": The paper's primary claim that the TPV captures the model's monitoring of its own reasoning process, is not sufficiently substantiated. The TPV is trained as a simple regressor mapping hidden states to a normalized token position (). It is highly probable that this regressor is not learning a high-level, semantic representation of "reasoning progress" but is instead capturing superficial, correlational text patterns. For example, certain phrases ("Let me double-check," "Wait," "The formula is...") naturally correlate with the beginning, middle, or end of a thinking sequence. The paper's own analysis in Appendix E, which shows that words like "okay" and "right" strongly and positively impact the progress value, seems to support this "text pattern" hypothesis far more than a "cognitive monitoring" one.
>
> We agree that distinguishing a genuine latent progress signal from a trivial “bag of phrases” explanation is important, and we thank the reviewer for highlighting this point. In response, we added a new section (“Semantic analysis of the TPV direction”) in the revised manuscript. There, we project the TPV direction onto the unembedding matrix and examine cosine similarities to (i) the 20 closest tokens and (ii) a curated set of “meta” tokens that might plausibly drive superficial pattern-matching (e.g., again, Hmm, Wait, problem, so, think, Double-check, Right, Okay, But, Hold on).
>
> The results strongly argue against a phrase-based explanation. The meta tokens show no consistent alignment with TPV: for example, “Okay” has a negative cosine similarity despite correlating positively with progress in Appendix E, while “But’’ and “Wait’’ show small positive similarities despite being associated with lower progress. The similarities are small and mixed in sign, indicating that TPV is not implementing rules like “if token X appears, increase progress.” Instead, TPV is a contextual direction that cannot be reduced to a small lexical dictionary.
>
> We also verify that TPV is not tracking proximity to the end-of-thinking token: cosine similarity(TPV, <\think>) is negative, ruling out the hypothesis that TPV approximates the likelihood of terminating the thinking phase.
>
> Moreover, the 20 closest tokens to TPV are almost entirely punctuation (e.g., “.”, “,”, “-”, “(”, “:”, “?”), all with very low cosine values (mean $\approx$ 0.029). Such tiny magnitudes show that TPV is not aligned with any meaningful token direction. Mild similarity to ubiquitous punctuation is consistent with TPV encoding a higher-level notion of movement through a sequence, not memorization of surface forms.
>
> Importantly, Appendix C further supports this interpretation: TPV emerges as a robust predictor of progress across all layers of the model, including early and mid layers that are far from the final layer where token-level semantics are expressed in the unembedding space. This layer-wise consistency is difficult to reconcile with a token-matching hypothesis but natural under a latent-process interpretation.
>
> Finally, our intervention experiments reinforce the causal role of TPV. Shifting hidden states along the TPV direction accelerates reasoning (more completions and more correct answers under a fixed token budget), while reversing the direction slows it down. These graded and reversible effects would not occur if TPV merely detected phrases like “Okay’’ or “Wait.’’ Together, the evidence indicates that TPV encodes a genuine internal progress signal rather than a shallow pattern over a few characteristic tokens.

---

> ### Author Response · Authors · 2025-11-22
> **Rebuttal by Authors (2/4)**
>
> > **W.2.** Fair Results of the "Overclocking" Intervention: Given the concern above, the success of the "overclocking" intervention feels tautological and less surprising. If the TPV  is simply a vector that correlates strongly with the end of a sequence, then applying an intervention  is, by definition, pushing the model's hidden state closer to a representation that is associated with sequence termination (such as < eos > token). It is therefore an expected outcome that this intervention shortens the output. This result does not prove that the model causally uses this vector for progress control, it only proves that the learned regressor has identified a manipulable direction in the activation space that correlates with sequence length.
>
> The intervention result is not tautological. **TPV is not trained on < eos > or any termination token**, it is trained only on hidden states inside the < think > … < /think > span. Therefore, TPV cannot be a learned “end-of-sequence direction.”
>
> While < eos > appears among the nearest neighbors of TPV in the unembedding space, its similarity is extremely small (≈0.03) and is on the same scale as punctuation and whitespace tokens such as “.”, “,”, “-”, and newline, which are tokens that clearly carry no semantic signal of termination. This pattern shows that TPV is not aligned with any meaningful token direction, let alone one that triggers sequence ending.
>
> Most importantly, if TPV simply pushed the model toward termination, we would expect premature truncation and lower accuracy. Instead, overclocking yields more completed trajectories and higher accuracy under the same token budget, and the effect scales smoothly with the intervention strength. This behavior is incompatible with nudging the model toward < eos > and is exactly what we expect from modifying an internal progress variable that governs the pace of reasoning rather than sequence termination.
>
> > **W.3.** Questionable Generalizability: The empirical evaluation is confined to two mathematical reasoning datasets (MATH-500 and GSM8K). This domain is highly structured and often procedural. It is highly questionable whether these findings would generalize to other, more common LLM tasks. For instance, how would a "progress vector" function in open-ended creative writing, complex document summarization, or dialectical tasks where the concept of "progress" is ambiguous, non-linear, or recursive? The paper provides no evidence or discussion on this crucial aspect of generalizability.
>
> We thank the reviewer for raising this and for the opportunity to strengthen our empirical results. In the revised version, **we extend monitoring experiments from two to five datasets**: in addition to Math500 and GSM8K (already discussed in Section 3.1), **we now include HumanEval (code generation), StrategyQA (flexible multi-hop commonsense reasoning), and GPQA (scientific reasoning)**. Across all five benchmarks, TPV achieves strong predictive performance with $R^2$ values between $0.61$–$0.75$ (Table 1). We also add an **intervention check on GPQA** (Table 4), where higher intervention strengths again increase both completed and correct trajectories under fixed token budgets, demonstrating that the overclocking effect extends beyond mathematics.
>
> These additions show that our method operates reliably across **diverse reasoning paradigms** such as procedural math problems, short multi-hop commonsense questions, code-generation tasks, and scientific reasoning queries, each with different structural characteristics. In contrast, domains such as open-ended creative writing or long-form summarization currently lack (i) standardized structured-reasoning benchmarks with explicit reasoning traces and (ii) clear, objective notions of task completion or correctness. Without these ingredients, it is not yet possible to define or evaluate a TPV-style progress signal in a meaningful way. We now clarify this distinction in the paper: TPV is designed for tasks where models exhibit an explicit, coherent reasoning phase and where progress can be operationalized; unconstrained generation tasks do not yet meet these conditions.

---

> ### Author Response · Authors · 2025-11-22
> **Rebuttal by Authors (3/4)**
>
> > **W.4.** Ambiguous Practical and Analytical Significance: From an interpretability perspective, the work does not provide strong evidence for the high-level claims of "metacognition" or "self-monitoring." It primarily demonstrates that text generation in a structured task has strong correlational patterns. From an engineering perspective, the method's value is not clearly demarcated from simpler alternatives. The "Instruct" baseline (a prompting technique) also performs well, and the best results are achieved when combining TPV with this prompting. This makes it unclear if this complex, model-specific intervention is practically superior to more robust methods like prompt engineering or fine-tuning with a length-based reward.
>
> A key contribution of our method is that it enables a real-time visual “loading bar’’ for reasoning, valuable both for interpretability research and for practical deployment, where millions of users interact with LLMs without any visibility into how far along the model is in its reasoning. We also disagree that our findings reduce to surface correlations or that prompting provides an equivalent alternative. Our analysis shows that TPV captures a genuine internal progress variable: it predicts progress consistently across layers (Appendix C), is nearly orthogonal to token semantics, and critically, its manipulation causally accelerates or slows reasoning in a graded and reversible manner. These effects cannot be produced by prompting. TPV also requires only minimal training data, leaves model weights untouched, and provides per-input control via the intervention magnitude $\alpha$, which fine-tuning or RL with length-based objectives cannot offer. Practically, TPV yields higher accuracy and more completed trajectories under identical token budgets, benefits that prompting or simple heuristics do not achieve.
>
> > **W.5.** Lack of Experimental Clarity: Key details of the methodology are ambiguous or omitted. The paper states that 30 problems were sampled, with 5 responses each, to train the regressor. This implies a training set of only ~150 trajectories, which seems exceptionally small for training even a linear regressor, let alone an RNN. The authors must clarify the exact dataset size (number of trajectories  and total data points ). It is also not explicitly stated whether the linear TPV or the RNN model was used for the main intervention experiments in Section 4.
>
> We thank the reviewer for highlighting this and for the opportunity to clarify the experimental setup. In the revised version, we now explicitly report the **dataset sizes**, the total number of hidden-state datapoints, in the newly added Table 1. As noted there, while we sampled 30 problems with 5 trajectories each, every trajectory contributes all of its decoding steps, resulting in tens to hundreds of thousands of training points per dataset, which is more than sufficient for learning the TPV regressor.
>
> Regarding which model is used for the main intervention experiments, Section 3.2 already specifies that the intervention operates by **projecting the hidden state along the learned TPV direction.** “... we conduct an intervention, which shifts the hidden representation h in the direction of projection vector θ …”  We emphasize this more clearly in the revision: all intervention experiments in Section 4 use the **TPV linear projection vector**, not the RNN-based smoother. The RNN is used only for producing a smoother visualization of the progress bar, not for intervention.
>
> > **Q.1.** The non-monotonic drops in progress (Fig. 5c, Fig. 9) are attributed to "hesitation or reflection." An alternative, and simpler, hypothesis is that the model has simply generated a token (e.g., "Wait") that the TPV regressor strongly associates with an earlier phase of problem-solving (a learned text pattern). How can the authors more rigorously disentangle this "text pattern" hypothesis from their "cognitive monitoring" hypothesis?
>
> This objection returns to the “text-pattern’’ explanation, which we have already directly tested, and disproven, via intervention results, TPV’s cross-layer predictiveness, and the new semantic analysis in the revised paper.
>
> > **Q.2.** The raw TPV predictions are clearly non-monotonic. The paper mentions using exponential smoothing and an RNN to create a smoother progress bar. How is the final progress bar visualization (as in Fig. 1a) rendered in real-time? Is it based on the smoothed or RNN-based output?
>
> All four variants are supported in our implementation: the raw linear TPV, its exponentially smoothed version, the lightweight RNN, and the exponentially smoothed RNN output. Any of these can be used to render a real-time progress bar. In practice, the RNN-smoothed variant provides the most stable and visually consistent signal, and this is the version shown in Fig.~1a.

---

> ### Author Response · Authors · 2025-11-22
> **Rebuttal by Authors (4/4)**
>
> > **Q.3.** Could the authors clarify the intervention mechanism? Is the intervention applied at every token generation step within the < think >...< /think > block? Furthermore, to understand the sensitivity of , what is the typical scale of the TPV's squared norm ()?
>
> Yes, the intervention is applied to any token generated (excluding the prefill) within the < think >...< /think > tokens. Additionally, following the reviewer’s suggestion, we report the norms of the TPV vectors in the revised manuscript; please see Appendix 7 (lines 835–850). Specifically, the norms of the TPVs range from 0.159 to 0.371 for DeepSeek-R1-Distill-Qwen-32B, and from 0.116 to 0.137 for DeepSeek-R1-Distill-LLaMA-8B.
>
> > **Q.4.** Could the authors comment on the hypothesized applicability and feasibility of this method for non-procedural, open-ended reasoning tasks, such as summarization or creative writing?
>
> Our method is designed for settings where the model exposes a structured reasoning phase, typically through an explicit < think > … < /think > segment. which is a dominant paradigm in current evaluations of LLM reasoning (math word problems, symbolic tasks, multi-step planning, scientific QA, etc.). These tasks are procedural by construction and produce a coherent internal reasoning trajectory, making them the natural domain for measuring and modulating “progress.”
>
> By contrast, tasks like summarization or creative writing do not involve a well-defined, stepwise solution process. They lack a canonical intermediate reasoning sequence, and the model’s generation is not aligned with a notion of “progress toward a solution” in the same sense. Because these tasks do not have a stable internal trajectory comparable to structured reasoning tasks, applying TPV directly would not yield a meaningful interpretation of progress.
>
> That said, the method is conceptually applicable whenever a model provides an explicit reasoning channel, either natively or through fine-tuning. In its current form, TPV is intended for structured reasoning tasks rather than open-ended generation, but the mechanism itself remains general and could extend to any setting where models produce a coherent, interpretable thinking stream.

---

### Official Review · Reviewer_pPTH · 2025-11-02

**Soundness:** 3
**Presentation:** 3
**Contribution:** 3
**Rating:** 6
**Confidence:** 4

**Summary:**

This paper presents an interesting study towards the internal evidence of the relative position signals within the explicit thinking phase. To achieve this purpose, the authors aim to learn a 'progress vector' that can project the hidden states into the estimated progress. Then, they further examine the possiblity of overclock and downclock the reasoning process. The experimental results well demonstrate the effectiveness of the notion 'progress vector' and progress intervention made via the former.

**Strengths:**

1. The idea of identifying the intrinsic mechanisms that encode a model’s relative position within its internal reasoning process is very interesting and useful.
2. The proposal of learning a 'progress vector' is simple yet effective.
3. The extensive expeirments well demonstrate the effectiveness of the proposed solution.

**Weaknesses:**

1. The authors only choose mathematical reasoning task for study. It is only a specific subarea of LLM reasoning. More reasoning tasks from other domains should be investigated.

**Questions:**

See the above Weaknesses.

---

> ### Author Response · Authors · 2025-11-22
> **Rebuttal by Authors**
>
> We thank the reviewers for their constructive feedback.
>
>
> > **W.1.** The authors only choose mathematical reasoning task for study. It is only a specific subarea of LLM reasoning. More reasoning tasks from other domains should be investigated.
>
>
> Thanks for the comment. Following your suggestions, we conducted **additional experiments in coding via HumanEval, commonsense reasoning via StrategyQA, and scientific reasoning via GPQA**. The results are presented in blue (see lines 292-299, 384-395 in the revised manuscript), and they show that TPV-based extraction of progress bar (TPV monitoring) can be applied robustly to other tasks (see Table 1 in the revised manuscript).
>
> Additionally, **the results in Table 4 of the revised manuscript demonstrate that TPVs can be used for overclocking and for accelerating reasoning on the GPQA task** with DeepSeek-R1-Distill-Qwen-32. In particular, with $\alpha = 10$, the overclocked model answers 15 more questions than the baseline and answers 5 more correctly under the 5k-token limit. Under the 10k-token limit, our model answers 12 more questions, with 4 of them answered correctly.
>
> For convenience, we attach the new tables below:
>
> **Table 1:** TPV monitoring performance, training/validation sizes (in thousands of data points), and resulting TPV norms for DeepSeel-R1-Distil-Qwen-32B (Q32B) and DeepSeel-R1-Distil-LLAMA-8B (L8B).
> | Dataset    | Train (k) Q32B | Val (k) Q32B | Train (k) L8B | Val (k) L8B | MSE Q32B | MSE L8B | R² Q32B | R² L8B | TPV Q32B | TPV L8B |
> |------------|----------------|---------------|----------------|--------------|----------|---------|---------|---------|-----------|-----------|
> | Math500    | 21.3           | 7.9           | 281.2          | 7.9          | 0.017    | 0.025   | 0.65    | 0.65    | 0.256     | 0.120     |
> | GSM8K      | 20.3           | 4.9           | 47.4           | 16.0         | 0.015    | 0.023   | 0.74    | 0.72    | 0.250     | 0.137     |
> | HumanEval  | 149.8          | 39.4          | 143.4          | 49.5         | 0.025    | 0.038   | 0.75    | 0.67    | 0.243     | 0.116     |
> | StrategyQA | 62.9           | 16.1          | 54.6           | 13.0         | 0.029    | 0.030   | 0.68    | 0.65    | 0.159     | 0.117     |
> | GPQA       | 479.6          | 128.1         | 404.1          | 163.2        | 0.035    | 0.032   | 0.61    | 0.65    | 0.371     | 0.126     |
>
> **Table 4:** TPV intervention on GPQA with DeepSeek-R1-Distill-Qwen-32B under two thinking budgets (5k and 10k tokens). We report the number of correct (\#Cr), answered (\#An) and ended (\#En) trajectories.
>
> | Method          | #Cr (5k) | #An (5k) | #En (5k) | #Cr (10k) | #An (10k) | #En (10k) |
> |-----------------|----------|----------|----------|-----------|-----------|-----------|
> | Base            | 54       | 70       | 67       | 71        | 97        | 98        |
> | TPV α = 5       | 56       | 69       | 66       | 70        | 99        | 97        |
> | TPV α = 10      | 59       | 85       | 83       | 75        | 109       | 110       |

---

> > ### Comment · Reviewer_pPTH · 2025-11-28
> > **The authors well address the questions.**
> >
> > Thanks for the feedback provided by the authors. The concern is well addressed, hence I will choose to raise the score further.

---

### Official Review · Reviewer_v9dA · 2025-11-02

**Soundness:** 1
**Presentation:** 2
**Contribution:** 2
**Rating:** 2
**Confidence:** 2

**Summary:**

This paper proposes "thinking progress vector" (TPV), a vector representation of what is claimed the encoding of its "interval reasoning process".
The paper uses linear models, MLPs, and RNN as tools to monitor the change of this state along the reasoning trajectory of LLMs.
The paper claims that intervening this vector can change the behaviour of LLMs and make it reason "faster", or what is called "overclocking" by the paper.
Empirical results on math reasoning tasks shows some evidence about the effect of intervening TPVs.

**Strengths:**

- The paper has some intuitive figures showing the discovered "thinking progress vector".

**Weaknesses:**

- The idea of estimating the progress of thinking as proposed in this paper does not make sense. It is beyond me why an LLM can maintain a state representing its reasoning progress if it does not know the correct answer to the problem even when the reasoning is just started. If an estimation of this really existed, then we would have asked this LLM to solve the halting problem, which has been mathematically proven to be impossible.

- The TPV might represent some characteristic tokens marking the steps in solutions to math problems. The paper unfortunately does not provide enough study along this direction. In lines 261-263, even the paper itself writes "a crucial question that arises is whether TPVs reflect a fundamental mechanism that the model uses to track its reasoning progress, or if they are merely residual artifacts that correlate with progress but do not play a causal role in the computation." this possibility is never ruled out in the rest of the paper.

- The concept of reasoning progress itself is poorly defined for real-world problems. What does reasoning progress mean for writing some code for a specific task? Does reasoning ends when the code is completed? Or When LLM finishes running it in with a python interpreter? What does reasoning progress mean for the travel sales person (TSP) problem, when it is impossible to solve in polynomial time?

**Questions:**

I do not have questions for this paper. I have not checked all the experiments of the paper, so I will put a 2 for my confidence.

I request that the paper be framed more accurately and align with the fundamentals of computer science in terms of the message it tries to convey.

---

> ### Author Response · Authors · 2025-11-22
> **Rebuttal by Authors (1/2)**
>
> Thanks for the comments. Below are our responses.
>
> > **W.1.** The idea of estimating the progress of thinking as proposed in this paper does not make sense. It is beyond me why an LLM can maintain a state representing its reasoning progress if it does not know the correct answer to the problem even when the reasoning is just started. If an estimation of this really existed, then we would have asked this LLM to solve the halting problem, which has been mathematically proven to be impossible.
>
>
> We believe this concern stems from a misunderstanding of what our method predicts. The TPV does not attempt to predict whether the model will arrive at a correct answer, whether the reasoning should halt, or whether the computation will succeed, nor does it attempt to determine how many steps the model should take. Instead, it estimates a relative position within the model’s own unfolding chain-of-thought, based on patterns in its hidden states.
>
> This is analogous to how a human solver can often tell roughly how far along they are in their reasoning, whether they are still setting up the problem, in the middle of working through a case, or wrapping up, without yet knowing the final answer. This sense of progress does not need to be perfect to be useful; it is simply a reflection of structure in how thoughts are organized.
>
> Indeed, tracking where we are in a reasoning process is widely recognized as an important component of being able to complete that process effectively [1]. These important findings in cognitive science make the lack of an analogous study for LLMs a clear gap in the current literature, which makes our work especially important.
>
> The TPV captures an analogous internal signal in LLMs. It is learned from the hidden-state trajectory of the finite, autoregressive reasoning segment that the model itself generates. The target is the normalized index within this bounded sequence. This is fundamentally different from predicting whether an arbitrary computation will halt, which is what the halting problem concerns. Our method does not, and need not, address anything beyond the model’s own generated thinking phase.
>
> In short, estimating “how far along the reasoning process I currently am” is a property empirically evident in reasoning models, just as it is for humans, and does not imply any ability to foresee correctness or solve undecidable problems.
>
> > **W.2.** The TPV might represent some characteristic tokens marking the steps in solutions to math problems. The paper unfortunately does not provide enough study along this direction…
>
> We agree that distinguishing correlation from causation is essential, which is precisely why the paper includes explicit intervention experiments to test whether movement along the TPV direction **causally** affects the model’s reasoning dynamics. During inference, we modify hidden states by adding scaled vectors in the TPV direction. If TPV merely captured characteristic tokens or other superficial correlations, such interventions would not reliably change the structure of the reasoning trajectory. Instead, we observe a clear causal effect: under several fixed token budgets, both the number of completed reasoning traces and the number of correct final answers increase (Tables 1 and 2). Moreover, our control experiments (Figure 6, Appendix B) show a monotonic and reversible pattern - positive $\alpha$ accelerates reasoning and negative $\alpha$ slows it. This behavior is incompatible with a purely correlational explanation and strongly suggests that TPV aligns with an internal mechanism used to regulate reasoning progress.
>
> We thank the reviewer for the opportunity to clarify this further. In the revised version, we added a new paragraph (see ‘Semantic Analysis of the TPV Direction’)  analyzing the TPV’s semantics in embedding space, which further rules out the “superficial correlation’’ hypothesis. TPV is nearly orthogonal to the very meta-tokens that would support such an explanation and instead corresponds to a latent direction that cannot be reduced to token-level patterns. Combined with the intervention results, this provides strong evidence that TPV encodes a genuine progress variable rather than an incidental correlation.

---

> > ### Author Response · Authors · 2025-11-22
> > **Rebuttal by Authors (2/2)**
> >
> > > **W.3.1.** The concept of reasoning progress itself is poorly defined for real-world problems. What does reasoning progress mean for writing some code for a specific task? Does reasoning ends when the code is completed?
> >
> > In this work, we follow the standard convention in reasoning-oriented LLMs, where reasoning refers to the autoregressive generation the model performs before producing its final answer. For models that employ an explicit thinking phase, which is the setting we study, this definition is stated clearly in Section 3.1. As noted in lines-155-159:
> >
> > “For a generation that ended successfully, we observe its thinking trajectory
> > $T = w_1 w_2 \dots w_N$, the sequence of tokens between < think > and < /think >.
> > We collect the final-layer representations $\{h_j^{(k)} \in \mathbb{R}^d \mid j = 1,\dots,N_k\}$ for each token in the trajectory.”
> >
> > This formulation makes explicit that reasoning progress is defined solely over the model’s internal, token-by-token thinking sequence between < think > and < /think > , not over the external task or its true solution process. In short, reasoning progress characterizes the model’s generation trajectory, not its progress toward solving the underlying problem.
> >
> >
> > > **W.3.2.** What does reasoning progress mean for the travel sales person (TSP) problem, when it is impossible to solve in polynomial time?
> >
> > As with any problem, for TSP the model can think for an arbitrary number of tokens, since it can allocate as many tokens as it wants between the < think > and < /think > markers (the explicit thinking phase). Indeed, TSP cannot be solved in polynomial time, but the model may terminate the thinking phase early and fail, succeed via heuristics, or produce a number of tokens that grows exponentially with the input size. In all of these cases, extracting a signal that estimates where the model is located during the thinking phase can still be highly informative.
> >
> >
> > > **W.3.3** Or When LLM finishes running it in with a python interpreter?
> >
> > As explained earlier, the reasoning ends when the model produces the < /think > token.
> >
> >
> > ___
> >
> > [1] R. Ackerman and V. A. Thompson. *Meta-Reasoning: Monitoring and Control of Thinking and Reasoning.*
> > Trends in Cognitive Sciences, 21(8):607–617, 2017. https://doi.org/10.1016/j.tics.2017.05.004

---

### Author Response · Authors · 2025-11-22
**General Response to Reviewer Comments**

It is well understood that human thinking involves an internal control mechanism that tracks the progress of the thinking process [1, 2]. Naturally, such a control signal, like many other estimates our brain makes, is not totally accurate. Given these well-accepted findings, our work, even just the monitoring component, fills an important gap in the study of LLMs as thinking machines.

We are fortunate to have curious reviewers who think for themselves. They raise important questions, but are naturally not expected, within the confines of the reviewing process, to explore the implications of these questions all the way through. For example, reviewer v9dA asks a very important question: if we know when a computation will end, does that not mean we solve the halting problem? The answer is that (1) ML predictions are just predictions, not definite answers, and (2) the LLM can terminate even if a “proper” computation would not.

Similarly, reviewer DLQv realizes that estimating the current location along the progress bar is equivalent, since we know the current autoregressive step, to estimating the length of the response. This is correct, but it does not diminish the significance of our finding; instead, it highlights its practical value, since knowing the expected length of ongoing reasoning is highly valuable both for users and for allocating computational resources.

Also relevant is reviewer VS5W’s concern that monitoring could simply arise from pattern matching. In the original manuscript, we show evidence to the contrary through our intervention experiments: shifting hidden states along the TPV direction causally accelerates reasoning (more completions and correct answers under fixed token budgets), while reversing the direction slows it down. These graded and reversible effects would not occur if TPV merely detected surface patterns. To further reinforce this point, we now add experiments that directly analyze TPV’s semantics in embedding space.

The depth and philosophical nature of the reviewers’ questions, about the halting problem, the nature of progress estimation, and the distinction between pattern matching and genuine cognitive mechanisms, reflect the fundamental importance of our work. By demonstrating that LLMs maintain internal progress signals during reasoning, we are touching on core questions about machine cognition that naturally invite such foundational scrutiny. We believe that all of these concerns, as well as other questions about the breadth of the work, are now fully addressed.

[1] R. Ackerman and V. A. Thompson. *Meta-Reasoning: Monitoring and Control of Thinking and Reasoning.*

Trends in Cognitive Sciences, 21(8):607–617, 2017. https://doi.org/10.1016/j.tics.2017.05.004

[2] Zhao, N., Teng, X., Li, W., et al. *A path model for metacognition and its relation to problem-solving strategies and achievement for different tasks.*

ZDM Mathematics Education 51, 641–653 (2019). https://doi.org/10.1007/s11858-019-01067-3

---

### Author Response · Authors · 2025-12-02
**Brief Summary of Rebuttal (1/2)**

Dear Area Chair and Senior Area Chair,

Thank you for overseeing our submission during a cycle complicated by OpenReview’s technical issues. To support an efficient final assessment, we summarize below the key reviewer concerns and our corresponding responses, together with the additional experiments and analyses we conducted during the discussion period.

**Reviewer v9dA**

| No. | Reviewer Concerns | Our Responses |
|-----|-------------------|---------------|
| **W1** | The idea of estimating “thinking progress” does not make sense; the model should not be able to track progress without knowing the correct answer. Reviewer argues this would imply solving the halting problem. | TPV does **not** predict correctness, halting, or success. It estimates **relative position within the model’s own finite chain-of-thought**, which is fundamentally different from the halting problem (which concerns unbounded computation). ML predictions are not guarantees, and LLMs may terminate even when an ideal algorithm would not. Cognitive science shows humans maintain internal progress signals during reasoning, and our work fills this gap for LLMs. |
| **W2** | TPV may simply detect characteristic tokens; causal interpretation not demonstrated. | **Causal interventions** produce graded speed-ups/slow-downs, which cannot arise from mere correlation. Semantic analysis further shows TPV is orthogonal to meta-tokens, ruling out pattern-matching. |
| **W3** | The concept of “reasoning progress” is poorly defined for tasks like coding, TSP, or interpreter execution. | Progress is defined only over the model’s internal `<think>`…`</think>` phase, not the external problem, consistent with standard reasoning-LLM setups. |

Reviewer v9dA raised largely conceptual concerns and explicitly noted low confidence. We provided detailed clarifications, new semantic analysis, and causal evidence. The reviewer did not engage further.

---

**Reviewer pPTH**
| No. | Reviewer Concerns | Our Responses |
|-----|-------------------|---------------|
| **W1** | Only mathematical reasoning was evaluated; broader domains should be investigated. | We added TPV monitoring on HumanEval, StrategyQA, and GPQA, all successful, **demonstrating strong generalization beyond math.** We also showed effective TPV intervention on GPQA, improving both answered and correct trajectories at 5k/10k budgets. |


Reviewer pPTH requested broader evaluation, which we conducted and included. They explicitly confirmed their concerns were resolved and increased their score.


---

**Reviewer VS5W**

| No. | Reviewer Concerns | Our Responses |
|-----|-------------------|---------------|
| **W1** | TPV may just be pattern-matching. | We added a **semantic analysis** showing TPV is nearly orthogonal to meta-tokens (“Wait”, “Okay”, “But”, etc.) and closest instead to low-magnitude punctuation directions, ruling out phrase-based mechanisms. Cross-layer predictiveness and causal interventions further confirm TPV reflects a real latent progress signal. |
| **W2** | Overclocking may merely push toward `<eos>`. | TPV is trained **only inside** `<think>... </think>`, never on `<eos>`. `<eos>` similarity is tiny (~0.03) and comparable to punctuation. Crucially, overclocking **increases accuracy and completion**, which contradicts the “premature termination” hypothesis. |
| **W3** | Generalization beyond math unclear. | We added results on HumanEval, StrategyQA, GPQA, demonstrating strong monitoring and intervention effects across diverse reasoning tasks. |
| **W4** | Limited practical significance vs. prompting. | TPV uniquely provides a **real-time progress bar**, **causal control** of the reasoning process, and **per-input adjustability** via α not achievable by prompting alone. |
| **W5** | Dataset size and intervention model unclear. | Table 1 now reports exact training/validation sizes (tens–hundreds of thousands of datapoints). All interventions rely on the **linear TPV vector**; the RNN is used only for smoothing the visualization. |
| **Q1** | Non-monotonicity may reflect token cues. | Addressed via semantic analysis, layer-wise robustness, and interventions. Drops correspond to real reasoning events (validation, re-evaluation), as shown in Appendices D and E. |
| **Q2** | Which smoother used in Fig. 1a? | All four variants supported; **RNN-smoothed** used for visual stability. |
| **Q3** | Intervention mechanism and TPV norm. | Applied every token inside `<think>...</think>`. TPV norms reported (0.159–0.371 for Qwen; 0.116–0.137 for LLaMA). |
| **Q4** | Applicability to open-ended tasks. | TPV applies where models expose an explicit reasoning phase. |

Beyond the reviewer’s initial acknowledgment of the strength of our mathematical-reasoning experiments, we addressed all concerns with new experiments (three new datasets, GPQA intervention), semantic analysis, clarification of methodology, and reporting of norms. The reviewer did not continue discussion after our detailed response.

---

> ### Author Response · Authors · 2025-12-02
> **Brief Summary of Rebuttal (2/2)**
>
> **Reviewer DLQv**
>
>
> | No. | Reviewer Concerns | Our Responses |
> |-----|------------------------------|--------------|
> | **W1** | Progress = relative position; predictor just learns total length; positional encodings make this trivial; no semantic signal. | Although relative position is mathematically equivalent total length, **this information is not present in positional encodings**, and its prediction is both useful and interpretable. TPV’s dynamic, non-monotonic behavior reflects real internal reasoning shifts. Appendix D shows drops exactly at validation phases, and Appendix E shows hesitation-token correlations **without embedding alignment**, confirming TPV tracks **semantic reasoning transitions**, not trivial position cues. |
> | **W2** | Math500 accuracy far below DeepSeek official numbers; setup “unreasonable.” | The gap stems from **multiple factors**: 32k vs. 2k–10k token limits; 420-problem test set; undisclosed DeepSeek-R1 system/prompt/decoding settings; output-format noncompliance (`\boxed{}`); and well-known **20–35% reproducibility swings** in R1 models as noted by multiple researchers. Our aim is **consistent improvements over our own baseline**, which TPV achieves. |
> | **Q1** | Compare natural generation vs. natural + TPV intervention. | We ran a **10k-token** evaluation. TPV (α=10) gives **+5.8% accuracy** and **~25% shorter** outputs, showing clear efficiency gains under realistic long-budget settings. |
> | **Follow-up Q** | Does non-monotonicity reduce usefulness? What does a drop mean? | Dynamic shifts are **expected** in reasoning. Appendix D shows drops occur at internal validation; Appendix E shows hesitation tokens correlate with drops, while TPV remains **orthogonal** to their embeddings. TPV reliably reflects **semantic transitions**, not noise or token matching. |
> | **Additional Concerns** | Prompt/sampling differences alone shouldn’t explain accuracy gap. | The discrepancy arises from the **combined** effects of token budgets, test-set differences, undocumented prompts/stopping logic, output-format issues, and known reproducibility variability, not prompt changes alone. |
>
>
> We provided two rounds of detailed clarifications and new experiments (long-budget runs, deeper semantic analysis).
>
> ---
>
>
>
> We hope this concise summary helps in making the final decision.
> We remain happy to provide any additional clarification.
>
> Sincerely,
>
> The Authors

---

### Meta-Review · Area_Chair_deeQ · 2026-01-06

**Summary:**

This paper investigates the reasoning phase regulation mechanism of Large Language Models (LLMs) with structured "thinking" tokens, proposing a vector representation named thinking progress vector (TPV) that encodes the interval reasoning process and reflects the model’s relative progress, and further employs linear models, MLPs and RNNs to track the TPV change along the LLMs’ reasoning trajectory, and introduces an "overclocking" intervention method that manipulates the model’s hidden states by adding a scaled TPV.

Most reviewers are of the opinion that this paper falls below the acceptance criteria of ICLR. The authors have provided detailed responses to the questions raised by the reviewers and addressed a portion of them. Having read this paper carefully, I consider it a work of certain interest. However, my primary concern is that the paper merely fits certain phenomena through experimental means, while the underlying essential causes of these phenomena are barely mentioned in the paper. Furthermore, I remain skeptical about whether these "findings" possess generalization ability. I hope the authors can conduct a more in-depth analysis of the causes behind the phenomena, instead of merely scratching the surface.

**Reviewer Concerns:**

I have perused this paper thoroughly, and I refrained from reading the reviewers’ comments so as not to compromise my objective assessment.

**Reviewer Scores:**

All the reviewers’ comments are relatively objective.

---

### Decision · Program_Chairs · 2026-01-26

Reject